# HLA matching or CRISPR editing of HLA class I/II enables engraftment and effective function of allogeneic human regulatory T cell therapy in a humanized mouse transplantation model

Oliver McCallion [1,10], Weijie Du[2,3,10], Viktor Glaser[2,3,4,10], Kate Milward [1], Sarah Short [1], Merve Bilici[1], Amy Cross[1], Helen Stark [1], Clemens Franke [2,3], Jonas Kath [2,3,5], Mikhail Valkov[2,4], Mingxing Yang[3], Leila Amini [2,3], Annette Künkele [2,6,7], Julia K. Polansky [3], Michael Schmueck-Henneresse [2,3], Hans-Dieter Volk [2,3,4], Petra Reinke [2,3,4], Dimitrios L. Wagner [2,3,4,5,8,9,11] ✉, Joanna Hester [1,11] ✉ & Fadi Issa [1,11] ✉

Regulatory T cells (Tregs) hold promise for treating autoimmune disease and transplant rejection, yet generation of autologous products for adoptive transfer can suffer donor variability and slow turnaround, limiting their use in urgent indications. We therefore examine whether allogeneic, pre-manufactured ('off-the-shelf') Tregs could overcome these barriers. In a human skin-xenograft model, HLA-mismatched Tregs are swiftly eliminated by recipient CD8$^+$ T cells and fail to protect grafts. Stringent matching of HLA class I and II restores efficacy but is clinically impractical. Using non-viral CRISPR editing we disrupt *B2M* and *CIITA* while inserting an HLA-E-*B2M* fusion, generating hypo-immunogenic Tregs that evade both T and NK cell attack. Engineered cells retain FOXP3 stability and potent in vitro suppression, and after a single low-dose infusion, prolong human skin graft survival in a humanized mouse model comparably to autologous Tregs. Histology and spatial transcriptomics reveal minimal cytotoxic infiltration and enrichment of immunoregulatory and tissue-repair programmes. Multiplex HLA engineering thus enables ready-to-use allogeneic Tregs that withstand host immune attack for adoptive transfer.

Regulatory T cells (Tregs) are potent immunosuppressive lymphocytes defined by expression of the FOXP3 transcription factor. Significant progress towards harnessing the therapeutic potential of Tregs to dampen pathological immune activity has been achieved with several recent successes in autoimmunity, graft-versus-host disease (GvHD), and transplant rejection[1–9].

To date, clinical trials have predominantly investigated adoptive cell transfer approaches. Here, cell therapy recipients donate blood or tissue from which Tregs are isolated, purified, and expanded ex vivo to create a bespoke infusion that is returned to the patient[10]. There are several challenges to this autologous approach. Firstly, individuals harbor variable numbers of Tregs, particularly at the

extremes of age, which potentially limits pre-expansion starting numbers and may restrict patients for whom autologous cell therapy is possible. Equally, the underlying pathology for which Treg therapy is required may be associated with impaired Treg function[11–14]. On the manufacturing side, Treg expansion takes several weeks to complete and may fail from biological or technical complications, including poor Treg expansion or contamination with non-Treg cells[15]. These manufacturing constraints preclude the acute use of autologous regulatory cell therapies, for example, in transplant recipients of deceased donor organs or during rejection episodes or autoimmune disease flare-ups. Finally, named-patient cell products require significant time, resources, and expertise to produce, reflected in significant manufacturing cost, which will likely limit their broad availability for patients[16].

For these reasons, the prospect of producing pre-manufactured cell therapy products from healthy human donors or a universal cell source, such as induced pluripotent stem cells (iPSCs), is appealing[17,18]. An "off-the-shelf" solution would facilitate selection and scaled production of maximally suppressive Tregs and would equally offer increased flexibility through providing a product available for acute, unscheduled administration, potentially enabling investigation of regulatory cell therapy for a broader range of pathologies[17]. Furthermore, the scaled manufacture of a cell product provides the ideal platform for engineering and banking of next-generation therapeutic products. However, as with any allogeneic cell therapy, it is likely that the recipient's immune system will reject infused cells, as already shown for allogeneic effector and CAR T cell therapies[19–21]. It is not fully understood yet whether this is also the case for unmatched Treg because of their inherent immunosuppressive capacity[22,23].

In this study, we develop methodologies that permit the in vivo survival and function of allogeneic Tregs. In vitro, allogeneic Tregs demonstrate similar suppressive capabilities to autologous Tregs, highlighting the difficulties interpreting in vitro functional assays. In vivo, killing by host CD8+ T cells dramatically reduces the efficacy of allogeneic Tregs, even when infused at relatively high numbers. This killing is circumvented by either HLA-matching of Tregs to the host or through the CRISPR-Cas9 silencing of HLA class I and II in the infused Treg cell therapy product. To further refine this approach, the replacement of polymorphic HLA class I with a non-polymorphic HLA-E-*B2M* fusion gene[24,25] restricts Treg elimination by natural killer (NK) cells. Together, these techniques define a set of successful strategies for effective "off-the-shelf" Treg cellular therapy.

## Results

### Allogeneic Tregs retain in vitro potency but lose in vivo efficacy

To compare the ability of Tregs to suppress either autologous or allogeneic responders, we assessed their function using standard in vitro suppression assays[26] in which suppression of proliferation of responder PBMCs is assessed in the presence or absence of Tregs. Autologous and allogeneic Tregs suppressed responder cell proliferation equally (Fig. 1A, B and Supplementary Fig. S1A–F). There were no significant differences in expression of the activation marker CD25 between responder cells suppressed by autologous compared to allogeneic Tregs (Fig. 1C, D). These results demonstrate that Treg suppress allogeneic and autologous effector cells equally well in vitro.

To determine the capacity for allogeneic Tregs to suppress the in vivo alloresponse, we utilized a mouse model of human skin transplant rejection[26] (Fig. 1E). Transplant survival was significantly prolonged by Treg treatment, although allogeneic Tregs demonstrated reduced efficacy in comparison to autologous cells (median graft survival time, MST, 81.5 vs >100 days) (Fig. 1F, G). This contrasted with in vitro suppression data, in which no differences were identified between the two Treg populations.

### CD8+ T cells eliminate allogeneic Tregs in vivo

To investigate the mechanisms underlying this reduced in vivo efficacy, we subjected Tregs to both in vitro and in vivo cell survival assays. First, autologous or allogeneic Tregs were cultured with freshly isolated PBMCs. Treg survival was calculated as a proportion of the initial seeding density. In this in vitro assay, no significant differences in Treg survival were identified (Fig. 2A). For the in vivo assay, unstained human PBMCs with or without CFSE-labeled Tregs were injected intraperitoneally into immunodeficient mice and recovered after 7 days by peritoneal lavage (Fig. 2B). Here, there was a significantly higher recovery of autologous compared with allogeneic Tregs (Fig. 2C), suggesting reduced in vivo survival of Tregs in an allogeneic host. To determine the host cells responsible for allogeneic Treg loss, we compared Treg survival in mice receiving total PBMCs or PBMCs depleted of either CD8+ or CD56+ cells (Supplementary Fig. S2B). Depletion of CD8+ T cells effectively restored the number and proliferative capacity of allogeneic Tregs recovered to a level comparable to autologous Tregs (Fig. 2D–F). Depletion of CD56+ cells did not have a similar impact, suggesting that the impaired survival and function of allogeneic unmodified Tregs in this model is predominantly driven by CD8+ cells.

To confirm the role of host alloimmunity on Treg survival, we investigated the effect of HLA-matching of Tregs with the host. We screened >150 Treg/PBMC pairs to identify matches between class I HLA (-A, -B, -C) and class II HLA (-DR). Three donor pairs were evaluated: B218-B209, B150-B208, and B130-B209 with HLA mismatches of (0,1,1,2), (0,0,0,1), and (1,2,1,2), respectively, across the 8 examined loci. This provided a range of partially matched Treg-recipient pairs (Supplementary Data 1). Skin allograft experiments were performed as before, but mice were treated with a reduced Treg dose to create a more challenging model[26]. Mice were treated with either partially matched or partially/completely mismatched allogeneic Tregs (Fig. 3A). While mice receiving completely and partially mismatched allogeneic Treg promptly rejected their grafts with an MST of 24 days and 27 days, respectively (Fig. 3B, C), most mice receiving partially matched allogeneic Treg developed long-term graft survival (>100 days). Two animals in the partially mismatched group developed xenoGvHD and were censored from the analysis from day 33, further illustrating the reduced efficacy of partially mismatched Tregs to control combined allo- and xeno- responses.

### CRISPR multiplex editing of HLA classes I and II creates hypoimmunogenic Tregs

Identification of HLA-matched donors for patients requires complex logistics, and complete matching remains a challenge[27,28]. Therefore, we explored whether targeted genetic modification of HLA may alleviate the need for stringent HLA-matching (Fig. 4). CRISPR-Cas9-mediated genetic disruption of *beta-2-microglobulin* (*B2M*) alone (II, *B2M* KO) and combined with deletion of *class II, major histocompatibility complex, transactivator* (*CIITA*) (III, double KO) eliminated the expression of HLA class I and class II, respectively (Fig. 4A–D). As *B2M*-edited T cells were shown to be targeted by NK cells via missing-self activation[29], we established a gene editing strategy to introduce the NK-cell inhibitory receptor HLA-E into Tregs. By integrating the coding sequence of the non-polymorphic *HLA-E* gene with a short linker sequence into the *B2M* exon 2, we installed an HLA-E-*B2M* fusion gene in HLA class I-negative Tregs (IV, HLA-E KI) (Fig. 4B–E). To further eliminate HLA class II in *B2M*-edited HLA-E knock-in (KI) Tregs, we silenced *CIITA* using a CRISPR-Cas9-derived adenine base editor (ABE)[30] in a second manipulation step (V, HLA-E KI/CIITA KO, H/C) (Fig. 4B–E). HLA-engineered Tregs displayed the canonical features of Treg identity, including a CD4+CD25+FOXP3+ phenotype (Fig. 4F), and no elevated Th1 cytokine production after polyclonal stimulation (Fig. 4G). All gene-edited Treg products suppressed the proliferation of allogeneic conventional T cells in a dose-dependent manner and

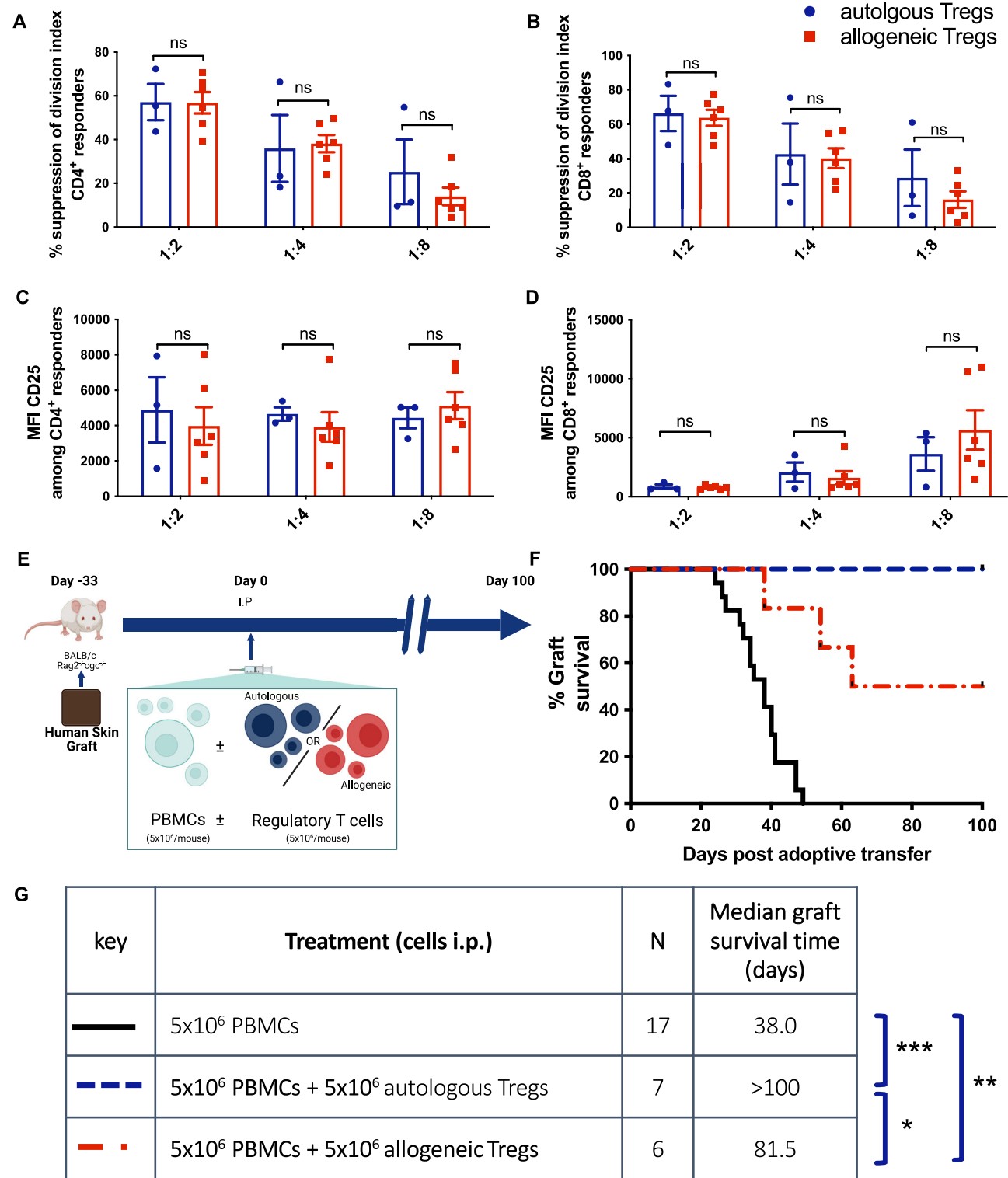

were comparable to autologous control Tregs in an in vitro suppression assay (Fig. 4H, I). Gene editing did not alter the low methylation state at the Treg-specific demethylation region (TSDR, Fig. 4J) in *FOXP3*.

## HLA-engineered Tregs evade NK-cell lysis

Fusing the non-polymorphic *HLA-E* cDNA with the endogenous *B2M* gene prevented surface expression of polymorphic HLA class I complexes in Tregs (Fig. 4B), but it should also reduce missing-self activation and cytolysis by endogenous NK cells[31]. To test this hypothesis,

we generated primary NK cell lines from healthy human donors and expanded them in vitro using cytokine-containing medium (Supplementary Fig. S3A, B). When co-culturing the pre-activated NK cells with Tregs, we observed an NK cell significant dose-dependent cytolysis of Tregs with deleterious edits at the *B2M* locus (Supplementary Fig. S3C). Analysis of the remaining Tregs confirmed dose-dependent lysis of HLA class I-negative Tregs (Supplementary Fig. S3D–F), but a preferential survival of HLA class I-positive and HLA class I-negative, HLA-E-*B2M* KI Tregs (Supplementary Fig. S3E–G). The relative fold-increase of HLA class I-positive Tregs was slightly higher than the

**Fig. 1 | Allogeneic Tregs match autologous cells in vitro yet protect skin grafts less effectively in vivo.** CFSE-stained human PBMCs ($1 \times 10^5$) were stimulated with αCD3/αCD28-coated beads ($2 \times 10^4$) with and without either autologous or allogeneic Tregs (at 1:2, 1:4, and 1:8 Treg:PBMC ratios) permuted from three donors mismatched at HLA-A, -B, and -DR loci (Suppl. Figure 1A). CFSE dilution after 72 h incubation was measured by flow cytometry, and a division index was calculated. **A** The percentage suppression of CD4$^+$ proliferation $F_{(1,6)} = 0.32$, $p = 0.59$, $n^2 p = 0.015$, 95% CI = [−10.23, 16.44], ($p > 0.05$ for all pairs). **B** The percentage suppression of CD8$^+$ proliferation $F_{(1,6)} = 1.03$, $p = 0.35$, $n^2 p = 0.04$, 95% CI = [−8.29, 20.24], ($p > 0.05$ for all pairs). Both are calculated relative to stimulated PBMCs cultured in the absence of Tregs. **C** Median fluorescence intensity of CD25 amongst CD4$^+$ responder cells $F_{(1, 6)} = 0.19$, $p = 0.68$, $n^2 p = 0.017$, 95% CI = [−148$^2$, 2120], ($p > 0.05$ for all pairs). **D** Median fluorescence intensity of CD25 amongst CD8$^+$ responder cells $F_{(1,6)} = 0.56$, $p = 0.48$, $n^2 p = 0.023$, 95% CI = [−2131, 1131], ($p > 0.05$ for all pairs). **A**–**D** Data are plotted as mean ± SEM of 3 autologous and 6 allogeneic responder:Treg donor combinations. All assays were performed in triplicate.

Statistical significance for autologous versus allogeneic Tregs, across all Treg:responder ratios, was assessed using two-way repeated measures ANOVA with two-tailed adjusted Bonferroni tests for pairwise comparisons. **E** Immunodeficient BALB/c Rag2$^{−/−}$ cγc$^{−/−}$ mice grafted with human skin received intraperitoneal human PBMCs ($5 \times 10^6$) alone ($n = 17$), with Tregs autologous to the PBMC donor ($5 \times 10^6$, $n = 7$) or Tregs allogeneic to the PBMC donor ($5 \times 10^6$, $n = 6$). Grafts were monitored for macroscopic signs of rejection over the following 100 days. Data are pooled from two independent experiments with each employing separate skin donors and distinct PBMC/Treg donor pairs. **F** Percentage of grafts surviving is plotted over time post-adoptive transfer of cells. **G** Sample size and median survival time for each treatment group is tabulated. Statistical significance was assessed using Mantel-Cox log rank tests: *$\chi^2(1) = 4.28$, $p = 0.0387$; **$\chi^2(1) = 11.69$, $p = 0.0006$, HR = 5.22, 95% CI = [2.17, 1$^2$.56]; ***$\chi^2(1) = 18.00$, $p < 0.0001$. Panel **E** created in BioRender. McCallion, O. (2025) https://BioRender.com/ynhfgt9. Source data are provided as a Source data file.

relative enrichment observed for HLA-E, indicating only partial protection from NK cell killing.

### HLA-edited Tregs promote graft survival in vivo

To test whether our HLA-engineered Treg cells are protected from allospecific T cell rejection, we established allospecific T cell lines by stimulating allogeneic CD56-depleted PBMCs with irradiated T cell-depleted PBMCs from our Treg donors as targets (Supplementary Fig. S4A). Prior to re-stimulation with allogeneic target cells, the alloreactive T cells were enriched for CD3$^+$ cells to remove contaminating NK cells and other CD3$^-$ cells (Supplementary Fig. S4B). The allospecific T cells comprised >90% TCRα/β$^+$ T cells with CD4$^+$ and CD8$^+$ T cell subsets and no NK cells (Supplementary Fig. S4B, C). Allospecific T cells induced dose-dependent lysis of the unmodified allogeneic Tregs (Supplementary Fig. S4G–H). In 3/5 allospecific T cell lines, we observed similar lysis of unmodified as well as allogeneic Tregs, which were only edited in the *B2M* locus (II-*B2M* KO, IV-HLA-E-*B2M* KI only) (Supplementary Fig. S4H). Tregs with silenced HLA class I and II were significantly better protected from allospecific T cell lysis than Treg edited at *B2M* alone (Supplementary Fig. S4I). Expanded Treg remained predominantly CD27$^{hi}$CD70$^{low}$ (Supplementary Fig. S4J). These data suggest that allospecific CD4$^+$ T cells can contribute to the rejection of unmatched allogeneic Tregs in an HLA class II-dependent manner.

We tested HLA-engineered Tregs in vivo under challenging conditions (Fig. 5A) with a complete mismatch (0/10 HLA-match) between donors for humanization and the Treg donor as well as the lower 1:5 Treg:PBMC ratio to better reflect clinical cell-dose constraints. To mimic a true "off-the-shelf" scenario, the Treg products were cryopreserved after manufacturing and thawed immediately prior to injection as previously described[32]. The phenotype of adoptively transferred cells is included in Fig. 4. As expected, allogeneic Tregs were unable to prevent the rapid rejection of transplanted human skin grafts (Fig. 5B). Fully edited allogeneic Tregs with HLA-E-*B2M* KI and *CIITA* KO (H/C Tregs) were detectable within peripheral circulation 21 days following adoptive transfer and promoted graft survival comparably to autologous control Tregs (Fig. 5B and Supplementary Fig. 5A–C).

Lastly, the skin transplantation model (Fig. 5A) was repeated to assess the effect of fully edited allogeneic H/C Tregs within the graft. In this experiment, skin grafts were harvested at day 15 following adoptive cell transfer. No graft rejection was observed following treatment with HLA-engineered allogeneic Treg (Fig. 5C). Spatial transcriptomic profiling demonstrated a strong rejection signature following treatment with either PBMC alone or with PBMC and allogeneic Treg, characterized by expression of transcripts involved in acute inflammatory responses, including inflammatory chemokine ligands and mediators of vasodilation and myeloid recruitment. This signature was

absent in grafts treated with either autologous or HLA-engineered Tregs, which instead exhibited transcripts associated with immune regulation (*CD5L*), lipid presentation (*CD1A*, *CD1E*), and graft homeostasis (*OGN*, *ARFGEF3*, and *PPP1R1B*) (Fig. 5D). Grafts treated with autologous or HLA-engineered Tregs exhibited restricted immune infiltration, including of cytotoxic T lymphocytes (Fig. 5E–G and Supplementary Fig. 5E), alongside preservation of cellular heterogeneity within the grafts (Supplementary Fig. 5F, G). Our results emphasize the necessity to silence or match both HLA class I and II to protect allogeneic Treg products from T cell immunity of mismatched donors.

### Discussion

Mounting evidence demonstrates the therapeutic efficacy of autologous Treg therapy in a variety of pathologies, mandating consideration of how this cell therapy approach may be scaled to deliver maximal benefit to all patient populations. To date, most clinical trials have been conducted with small numbers of patients in academic-sponsored efforts with manufacture of cell products by small GMP-compliant in-house units[4]. For example, manufacture of the autologous Treg product for our on-going phase 2b trial involves numerous manual steps undertaken by highly trained scientists[10,33]; simply put, to increase the number of products produced in this facility would require a linear increase in scientists, facilities, and equipment. From an economic perspective, therefore, there is currently limited potential to reduce costs by scaling up production. This may significantly hamper transition of the technology into the pharmaceutical industry and increase the risk of traditional funding institutions rejecting the approach in its entirety. Beyond the technical elements, the unpredictability inherent in producing a biological "living" product introduces significant variability in both the input and output of what is otherwise a standardized manufacturing process. Moreover, Treg cell therapy production is carried out over several weeks, making rapid treatment with an autologous product impossible for newly diagnosed conditions (e.g., new onset diabetes) or where the timing of treatment cannot be planned (e.g., deceased donor organ transplantation). Several clinical trials of regulatory cell therapy have been prematurely terminated due to manufacturing difficulties with the cellular product[34].

Here, we have demonstrated that Tregs suppress in an HLA-agnostic fashion, with equal potency towards autologous and allogeneic effector cells in vitro. Unlike conventional effector T cells, allogeneic Tregs do not result in the development of GvHD in vivo[35,36]. As expected, allogeneic Tregs were subject to CD8$^+$-mediated depletion in vivo, which effectively reduced the number of cells available to exert an immunosuppressive effect. Depletion of CD8$^+$ T cells but not CD56$^+$ NK cells restored survival and the immunoregulatory potency of unmatched allogeneic Tregs in vivo by restoring both total and undivided allogenic Treg counts to autologous levels, although PBMC

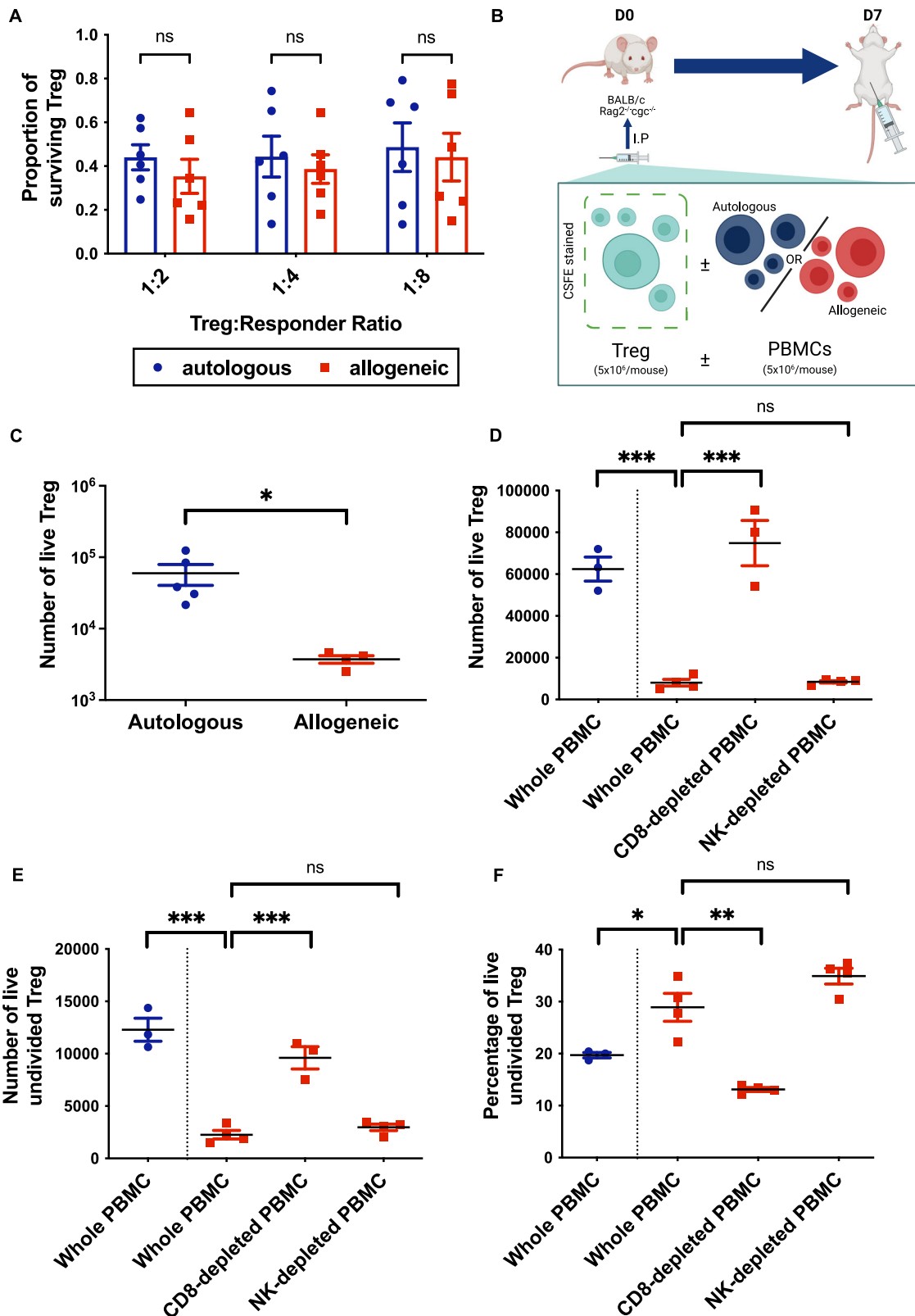

division was unable to be quantified as Treg alone were labeled, and subtle NK contributions to Treg survival may not be detectable. A discrepant response between in vitro and in vivo CD8⁺ T cell depletion is notable in our data and may reflect differential co-stimulation or cytokine signaling present in vivo. Additional mechanistic work to fully evaluate this observation may be valuable. Nevertheless, by creating an immunologically inert environment at the time of Treg administration,

either by stringent HLA-matching (7/8) or gene editing of HLA class I and II, we demonstrate that Tregs survive for long enough to establish a regulatory environment, conferring a similar degree of allograft tolerance to autologous cells. These findings provide proof-of-concept that HLA matching and/or gene editing can restore the in vivo efficacy of allogeneic Tregs to levels comparable with autologous cells under

**Fig. 2 | Allogeneic Treg survival is impaired in vivo and reversed by depletion of host CD8⁺ cells. A** Human ex vivo-expanded Tregs were cultured with VPD-stained autologous or allogeneic PBMCs (1 × 10⁵) in the presence of αCD3/αCD28-coated beads at Treg:PBMC ratios of 1:2, 1:4, and 1:8. After 3 days, live Tregs were quantified by flow cytometry and plotted as a fraction of the input number. Data are presented as mean ± SEM. Six Treg donors were used across conditions. Statistical analyses used a two-way repeated measures ANOVA with Bonferroni post-tests: $F(1,15) = 2.00$, $p = 0.18$; all pairwise comparisons $p > 0.05$. **B** Immunodeficient BALB/c Rag2⁻/⁻cγc⁻/⁻ mice received intraperitoneal CFSE-labeled human Tregs (5 × 10⁶) with 5 × 10⁶ autologous ($n = 5$) or allogeneic ($n = 4$) unstained PBMCs from one (autologous) or two (allogeneic) donors. After 7 days, peritoneal cells were recovered by lavage, and CD4⁺CFSE⁺ Tregs enumerated by flow cytometry. **C** Treg numbers/mouse plotted with mean ± SD for each treatment group. Mann–Whitney $U = 0$, $p = 0.0159$ (two-tailed). **D–F** Immunodeficient BALB/c Rag2⁻/⁻cγc⁻/⁻ mice received intraperitoneal CFSE-labeled human Tregs (5 × 10⁶) with 5 × 10⁶ autologous ($n = 3$) or allogeneic ($n = 11$) PBMCs from one (autologous) or two (allogeneic) donors (Supplementary Fig. 2A). Allogeneic PBMCs were either whole ($n = 4$), depleted of CD8⁺cells ($n = 3$), or depleted of CD56⁺ cells ($n = 4$). Depletions were

performed using magnetic beads. Cells were analysed on day 7. Treg numbers are shown/mouse with group mean ± SD. **D** Absolute Treg numbers. Statistical significance was calculated using two-tailed t-tests: autologous versus allogeneic PBMCs $t(5) = 10.50$, $p = 0.0001$, $n^2 = 0.96$, 95% CI = [−67700, −41000]; whole versus CD8⁺-depleted PBMCs $t(5) = 7.20$, $p = 0.0008$, $n^2 = 0.91$, 95% CI = [43000, 91000]; and whole versus CD56⁺-depleted allogeneic PBMCs $t(6) = 0.2393$, $p = 0.8188$, $n^2 = 0.009$, 95% CI = [−3674, 4470]. **E** Absolute number of undivided Tregs. Statistical significance was calculated using two-tailed t-tests: autologous versus allogeneic PBMCs $t(5) = 9.60$, $p = 0.0002$, $n^2 = 0.95$, 95% CI = [−13000, −7000]; whole versus CD8⁺-depleted PBMCs $t(5) = 7.20$, $p = 0.0008$, $n^2 = 0.91$, 95%CI = [4700, 10000]; and whole versus CD56⁺-depleted allogeneic PBMCs $t(6) = 1.38$, $p = 0.22$, $n^2 = 0.24$, 95% CI = [−550, 2000]. **F** Percentage of CFSEʰⁱ (representing undivided) Tregs. Statistical significance was calculated using two-tailed t-tests: autologous versus allogeneic PBMCs $t(5) = 2.89$, $p = 0.034$, $n^2 = 0.63$, 95% CI = [−17, −1]; whole versus CD8⁺-depleted PBMCs $t(6) = 5.9$, $p = 0.0011$, $n^2 = 0.85$, 95% CI = [9.21, 22.4]; and whole versus CD56⁺-depleted allogeneic PBMCs $t(6) = 2$, $p = 0.0983$, $n^2 = 0.39$, 95% CI = [−13.5, 1.51]. Panel B created in BioRender. McCallion, O. (2025) https://BioRender.com/35ntrd5. Source data are provided.

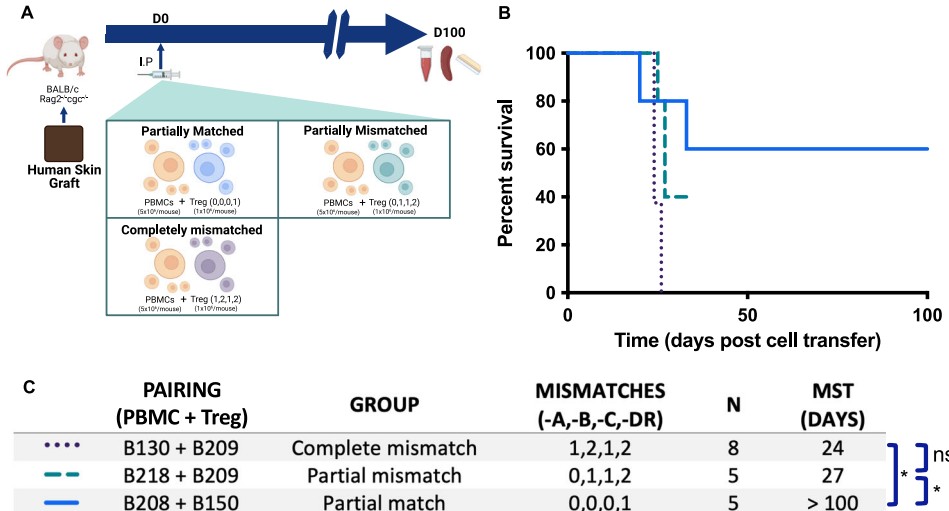

| | PAIRING<br>(PBMC + Treg) | GROUP | MISMATCHES<br>(-A,-B,-C,-DR) | N | MST<br>(DAYS) | |
|---|---|---|---|---|---|---|
| •••• | B130 + B209 | Complete mismatch | 1,2,1,2 | 8 | 24 | ns |
| – – – | B218 + B209 | Partial mismatch | 0,1,1,2 | 5 | 27 | * |
| —— | B208 + B150 | Partial match | 0,0,0,1 | 5 | > 100 | * |

**Fig. 3 | Near-complete HLA matching restores long-term graft protection by allogeneic Tregs. A** Immunodeficient BALB/c Rag2⁻/⁻ cγc⁻/⁻ mice grafted with human skin received intraperitoneal human PBMCs (5 × 10⁶) and either completely mismatched (1,2,1,2, $n = 8$), partially mismatched (0,1,1,2, $n = 5$), or partially matched (0,0,0,1, $n = 5$) Tregs (1 × 10⁶) at HLA-A,-B,-C,-DR loci identified through screening over 150 distinct donors. Grafts were monitored for macroscopic signs of rejection over the subsequent 100 days. **B** Percentage of grafts surviving is plotted over time post-adoptive transfer of cells. The mice receiving partially mismatched Tregs developed signs of xenogeneic graft-versus-host-disease necessitating

removal of these animals from the experiment at day 33. **C** Tabulated median survival time (defined as the time at which half of the grafts were rejected). Groups were analyzed using the two-tailed Mantel-Cox log-rank test with multiple testing correction using the Benjamini–Hochberg method ($\chi^2(2) = 7.84$, $p = 0.0198$); complete mismatch versus partial mismatch $\chi^2(1) = 6.021$, $p = 0.0141$; complete mismatch versus partial match $\chi^2(1) = 4.376$, $p = 0.0364$; partial mismatch versus partial match $\chi^2(1) = 0.3254$, $p = 0.5684$). Panel **A** created in BioRender. McCallion, O. (2025) https://BioRender.com/re81qq1. Source data are provided as a Source data file.

an off-the-shelf manufacturing workflow, with further work necessary to establish a scalable production strategy.

Allogeneic Tregs may exert their immunomodulatory function through TCR-dependent and independent pathways. TCR/HLA-independent suppression likely includes expression of co-inhibitory molecules such as cytotoxic T lymphocyte antigen (CTLA-4), production of immunomodulatory cytokines such as TGF-β, IL-10, IL-35, preferential utilization of IL-2 through expression of the high-affinity IL-2 receptor, and induction of apoptosis in effector populations through granzyme expression[2]. Both upstream expression of inhibitory ligands or the production of inhibitory cytokines, and their respective downstream receptors, are HLA-independent and therefore ex vivo Treg activation alone would be expected to promote an immunosuppressed milieu. Further, the Treg population contains cells with alloreactive T cell receptors[37], which could boost immunomodulatory function, given that such alloreactive Tregs were shown to be preferentially activated in the allograft and draining lymph nodes in mice

models[38,39]. As the frequency of alloreactive Tregs is typically low (<10%), future studies should evaluate the inclusion of chimeric antigen receptors to increase the frequency of (allo-)antigen-specific Tregs in the cell product to enhance their efficacy[40,41]. The development of hypoimmunogenic, gene-edited 'off-the-shelf' cellular therapies has been proposed for tissue replacement[24,42], anti-tumor cell therapy with conventional T cells[25,43–45], and now, allogeneic Treg applications. Silencing of HLA on allogeneic cells may avoid product-specific sensitization and allow redosing of allogeneic cell products in case of relapse or for consolidating treatment[46]. Introduction of NK-cell inhibitory receptors will be required to improve the persistence of allogeneic Treg with disrupted HLA class I expression[46]. We demonstrate that an HLA-E-*B2M* fusion gene allows partial protection against NK cells. HLA-E is a non-classical HLA class Ib molecule with only two alleles present in diverse populations (HLA-E*01:01 and HLA-E*01:03)[47]. HLA-E primarily presents signal peptides from other HLA-I molecules; however, some reports also demonstrate that HLA-E*01:03 can present

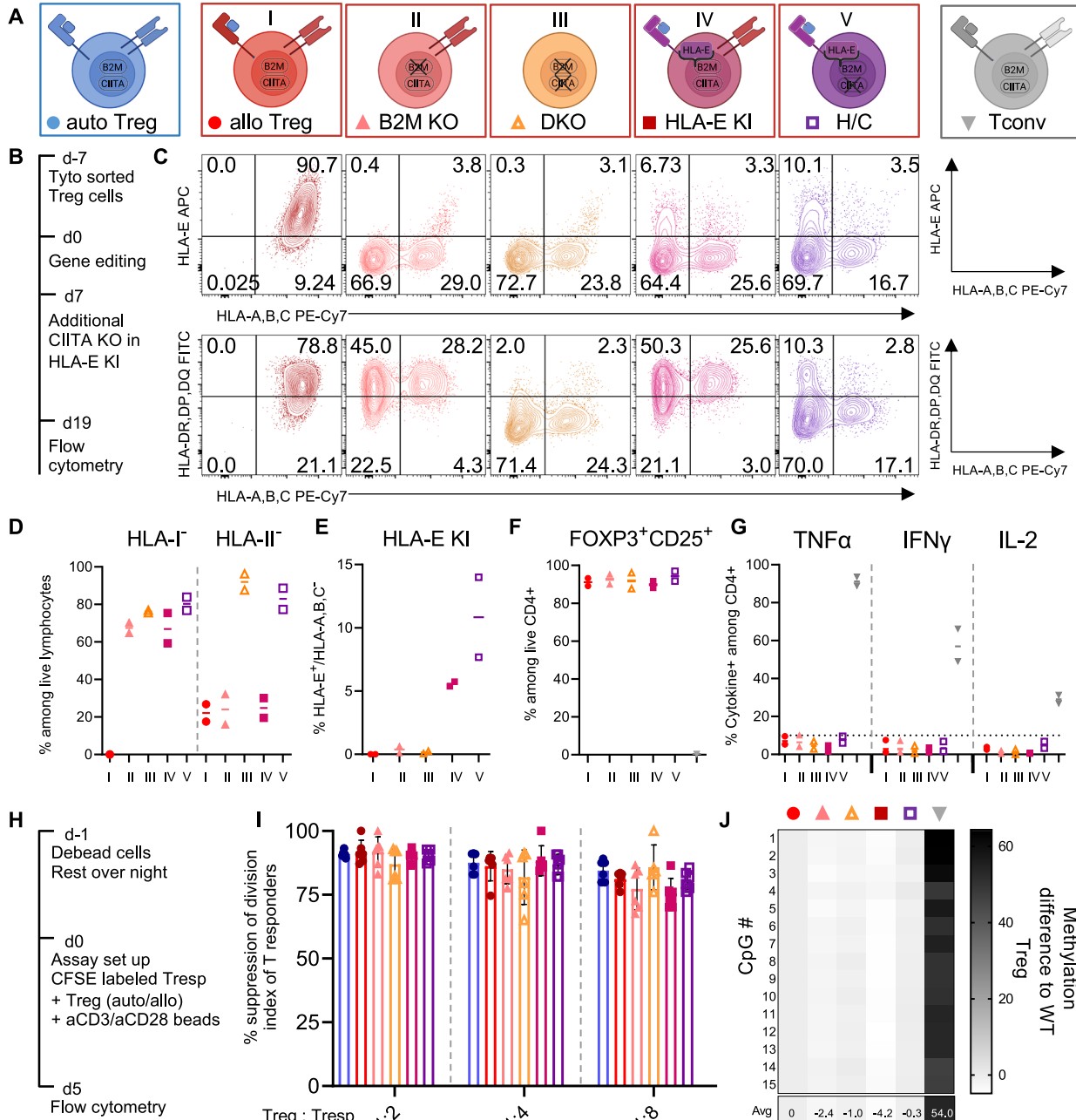

**Fig. 4 | Multiplex CRISPR editing generates Tregs that retain canonical phenotype and function. A** Gene editing strategy using CRISPR/Cas9 to disrupt HLA class I and class II by targeting *B2M* and/or *CIITA*, respectively. HLA-E was inserted with a short linker sequence into *B2M* exon 2, generating an HLA-E-*B2M* fusion protein. For H/C Tregs the additional *CIITA* KO was performed by splice site disruption using the adenine base editor ABE-8.20-m. **B** Timeline of cell isolation, gene-editing, and surface protein expression measurement. **C** Representative contour plots of editing outcome by HLA-A,B,C (PE-Cy7), HLA-DR,DP,DQ (FITC), and HLA-E (APC) staining. **D** Summary of editing outcome for two biological replicates showing frequency of HLA class-I and class-II negative cells and **E** of HLA-E positive and HLA-A,B,C negative knock-in cells among live lymphocytes. **F** Treg identity was confirmed by staining of CD25 and intracellular staining of FoxP3, as well as **G** intracellular cytokine (TNFα, IFNγ, IL-2) expression after PMA/Ionomycin stimulation with conventional T cells as controls. **H** Timeline of suppression assay: CFSE-stained human CD3+ cells ($2.5 \times 10^4$) with and without either autologous or the different allogeneic Treg conditions (at 1:1, 1:4, and 1:8 Treg:Tresp ratios) were stimulated with αCD3αCD28-coated beads at a 1:1 ratio of beads to total cells (Treg +Tresp). **I** CFSE dilution after 5 days incubation was measured by flow cytometry, and a division index was calculated. Data are presented as mean values ± SD. **J** Targeted DNA-methylation analysis of the 15 CpGs within the *FOXP3*-TSDR using bisulfite amplicon sequencing. Data were normalized by subtracting the methylation frequency of the WT Treg sample. Mean of two biological replicates is shown. Panel A Created in BioRender. Wagner, D. (2025) https://BioRender.com/ydoiabq. Source data are provided as a Source data file.

viral peptides from CMV or EBV[48,49]. Binding of HLA-E to CD94/NKG2A or CD94/NKG2B on NK cells or CD8 T cells inhibits said cells, explaining the protective effect observed in vitro[50]. In contrast, HLA-E can also present viral peptides to activate NK cells by binding the conserved NKG2C[51] or EBV-specific and HLA-E-restricted CD8+ T cells[48,52]. Previous strategies to create hypoimmunogenic cells

implemented HLA-E fusion proteins that lack antigen-presentation by covalently linking B2M, HLA-E, and a non-polymorphic signal peptide of HLA-G into a single molecule[24,25]. Our HLA-E-*B2M* fusion gene is based on HLA-E*01:03 without a blocking peptide and therefore may retain some ability to present pathogenic peptides. The severely compromised antigen presentation in HLA-edited Treg could provide

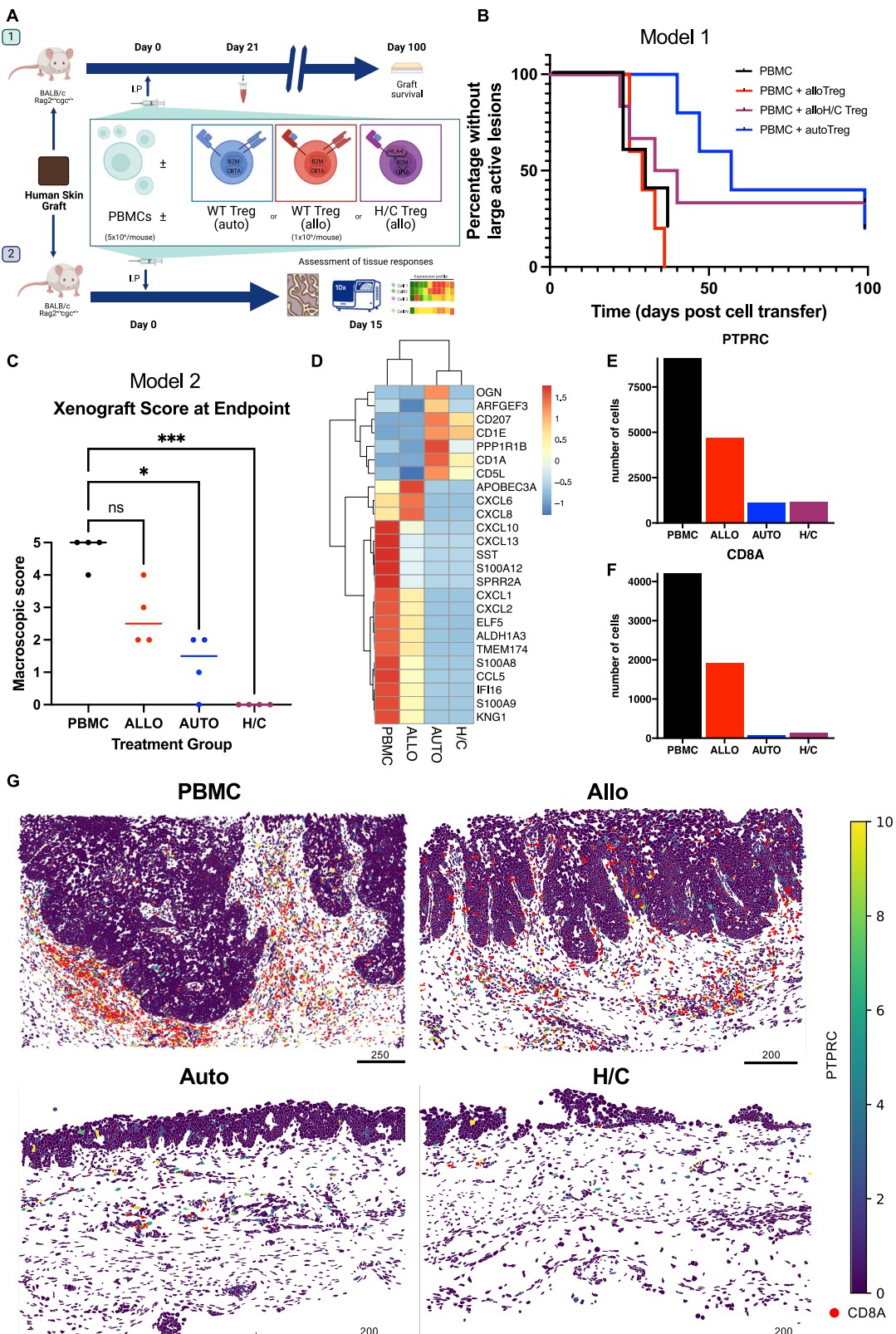

a reservoir for intracellular pathogens or malignant transformation, which may not be readily detected by T cell mediated immuno-surveillance. Therefore, the first clinical application of HLA-engineered hypoimmunogenic Treg cell product may warrant inclusion of a genetic safety switch.

Allogeneic Treg therapy requires installation of multiple edits to reduce immunogenicity of HLA class I and II in ex vivo expanded Tregs

from healthy human donors. The latter requires effective gene editing at multiple sites. Targeting multiple genes with conventional CRISPR-Cas9 provokes high rates of translocations with unknown biological function[53,54]. To overcome this, we separated the editing step between KI of HLA-E and silencing of *CIITA* to eliminate HLA-II with an ABE enzyme. Base editors, such as the employed ABE[30], introduce targeted base changes without inducing DNA breaks and thereby reduce the

**Fig. 5 | HLA-engineered Tregs promote long-term protection of human skin transplants to a degree comparable to autologous Tregs. A** Immunodeficient BALB/c Rag2$^{-/-}$ cγc$^{-/-}$ mice grafted with human skin received intraperitoneal human PBMCs (5 × 10$^6$, $n = 5$) alone, with Tregs autologous to the PBMC donor (1 × 10$^6$ Treg, $n = 5$), with Treg allogeneic to the PBMC donor (1 × 10$^6$ Treg, $n = 5$), or with allogeneic HLA class 2 KO and HLA-E KI Treg (1 × 10$^6$, $n = 6$). Grafts were monitored for macroscopic signs of rejection over the following 100 days. **B** Percentage of grafts without large active lesions is plotted over time post-adoptive transfer of cells. Statistical significance was assessed using log rank test for trend ($\chi^2(1) = 4.089$, $p = 0.0432$). **C–E** Immunodeficient BALB/c Rag2$^{-/-}$ cγc$^{-/-}$ mice grafted with human skin received intraperitoneal human PBMCs (5 × 10$^6$) either alone ($n = 4$) or with autologous ($n = 4$), allogeneic ($n = 4$), or H/C edited ($n = 4$) Tregs (1 × 10$^6$). Grafts were monitored for macroscopic signs of rejection and were harvested at 15 days following adoptive transfer. **C** Macroscopic graft rejection score at study end.

Individual data points with median group values are plotted. Groupwise differences assessed with the Kruskal-Wallis test against the reference group of PBMC (H(3) = 13.28, $p = <0.0001$, $\epsilon^2 = 0.86$). Post-hoc groupwise two-tailed Dunn's tests: (allo) $Z = 1.3$, $p = 0.195$, $r = 0.32$; (auto) $Z = 2.40$, $p = 0.016$, $r = 0.60$; (H/C): $Z = 3.47$, $p = 0.0005$, $r = 0.87$). **D–G** Spatial mapping of 427 transcripts across five skin grafts (one each from experimental groups treated with PBMC alone, autologous Treg, or allogeneic Treg, and two from the experimental group treated with H/C Treg) **D** Heatmap of $z$ values for the top 25 differentially expressed genes within skin grafts at study end. **E** Number of cells expressing *PTPRC* in skin grafts. **F** Number of cells expressing *CD8A* in skin grafts. **G** *CD8* and *PTPRC* expression across grafts by experimental group. Scale bars represent pixel values, where 1px represents 0.2125 μm. Panel **A** Created in BioRender. McCallion, O. (2025) https://BioRender.com/a0dop2t. Source data are provided as a Source data file.

risk of gross genetic rearrangements in the gene edited cell product. In future, our editing strategy may be adapted to enable multiplex-editing in a single manipulation to reduce manufacturing using different nucleases[55] or novel gene editing tools for DSB-free KI[56,57] when requiring integration of additional transgenes, such as a CAR[40,41]. Regardless of the editing strategy employed, there is a high likelihood for residual expression of HLA on the alloTreg surface or incomplete HLA-E expression. Sensitization against the Treg donor's alloantigens represents a potential limitation for redosing as well as potential clinical side-effects (e.g., for later transplantation of an organ/tissue expressing the Treg donor's HLA haplotype). Therefore, an optimized manufacturing approach should include a purification step to remove residual HLA class I/II whenever warranted. Further work should also examine whether lymphokine-activated unspecific killing is a barrier. Lastly, the experimental conditions presented here may be more challenging than the environment ultimately faced by HLA-engineered allogeneic Treg in clinical practice, given that recipients may not be completely immunocompetent. As such, evaluating the combination of HLA-engineered Treg and low-dose conventional immunosuppression is an important area of future study.

Allogeneic Tregs offer a promising therapeutic approach, but their immunosuppressive capacity is hindered by the anticipated alloresponse. Here, we pave the way for two viable strategies to enable their effective use. The first approach involves the use of HLA-matched allogeneic Tregs, while the second strategy incorporates unmatched Tregs with HLA class I and class II KO, alongside the introduction of HLA-E. Our results demonstrate that both of these approaches improve the therapeutic efficacy of allogeneic Tregs, presumably by extending their survival under alloantigen-specific immune pressure, although long-term persistence beyond 3 weeks is challenging to determine in our in vivo models. The advanced HLA-matching (>6/8 HLA matched) required for improved allogeneic Treg performance represents a significant logistical challenge and is impractical for a true 'off-the-shelf' approach. Consequently, HLA-editing provides a more tenable solution for clinical translation in the immediate future. Future work may include more in-depth studies to enhance the persistence of the HLA-edited Tregs and compare the HLA-E-transgene with other strategies to prevent NK-cell rejection[58–60].

This study, therefore, presents a significant advancement towards the clinical application of "off-the-shelf" Treg cell therapy derived from allogeneic donor cells. While this study focuses on allogeneic settings, the mechanisms by which HLA-edited Tregs resist rejection may also be beneficial in other immune-mediated diseases. Future work will determine their applicability beyond transplantation.

## Methods

All relevant ethical regulations have been adhered to in the production of this work. Ethical approval was provided by the Oxfordshire Research Ethics Committee (REC B), study number 07/H0605/130, and Charité (Berlin, Germany) ethics committee, study number EA4/091/19.

### Cell isolation and cell culture

Human peripheral blood mononuclear cells (PBMC) were isolated from the blood of healthy donors (provided by the National Blood Service, Oxford, UK) or was obtained after informed and written consent (Charité ethics committee approval EA4/091/19) using Lymphosep (Biowest, Nuaillé, France) for density gradient centrifugation. Erythrocytes were lysed using PharmLyse buffer (BD biosciences, Franklin Lakes, NJ, USA). Pre-enrichment of CD25$^+$ cells from PBMCs was performed using CD25 microbeads (Miltenyi Biotech, Bergisch Gladbach, Germany) according to the manufacturer's protocol. Depletion of CD8$^+$ cells or CD56$^+$ cells from PBMCs was performed using αCD8 or αCD56 microbeads (Miltenyi Biotech) according to the manufacturer's protocol. Treg (CD4$^+$CD25$^+$CD127$^{lo}$) were FACS-sorted from CD25-enriched PBMCs using BD FACSAria (BD Biosciences) after staining with αCD4−ECD (Beckmann Colter, Brea, CA, USA), αCD25−PE-Cy7 (BD Biosciences), and αCD127−PE (BD Biosciences). CD4$^+$ effector T cells (Teffs) were isolated from CD25-depleted PBMCs using αCD4 microbeads (Miltenyi) according to the manufacturer's protocol. Cells were cryopreserved in freezing medium comprising 50% heat-inactivated FCS and 50% RPMI1640 with 10% DMSO. Prior to assay cells were thawed in a 37 °C water bath before dilution of DMSO with RPMI1640 warmed to 37 °C.

For experiments intended for gene editing, pre-enrichment of CD4$^+$ cells from PBMCs was performed using CD4 microbeads (Miltenyi Biotech) according to the manufacturer´s protocol. Due to their higher stability[61], CCR7$^+$ Tregs (CD4$^+$CD25$^+$CD127$^{low}$CCR7$^+$) were sorted using a Tyto sorter (Miltenyi Biotech) after staining with αCD4 VioBlue (Miltenyi, REAL103), αCD25 APC (Miltenyi, REAL128), αCD127 PE-vio770 (Miltenyi, REAL102), αCD45RA FITC (Miltenyi, REAL164) and CCR7 PE (BioLegend, G043H7). Sorted Tregs were cultured in a 96 U well plate with 100,000 cells in 200 μl Treg medium per well. Treg medium consists of X-Vivo 50 (Lonza) medium supplemented with 10% heat-inactivated FCS, 500IU/mL of recombinant human interleukin-2 (IL-2) (Miltenyi, Bergisch Gladbach, Germany) and 100 nM Rapamycin (Pfizer). T cells were cultured in T cell medium (RPMI1640 supplemented with 10% FCS, 10 ng/mL IL-7, and 5 ng/mL IL-15). NK cells were enriched from PBMCs using the NK isolation Kit (Miltenyi) and cultured in NK MACS Medium (Miltenyi) supplemented with 10% FCS, IL-2 (500IU /mL) and IL-15 (5 ng/mL). The K562 cell line was cultured in RPMI1640 supplemented with 10% FCS.

For in vitro cell cultures and assays, leukocytes were cultured in RPMI1640 supplemented with L-glutamine (Sigma-Aldrich, St. Louis, MO, USA), 100 U/mL penicillin and 10 mg/mL streptomycin (Sigma-Aldrich), and 10% heat-inactivated human serum (Seralab, Haywards Heath, U.K).

### Treg expansion

After isolation, Tregs were cultured for 16 days in complete medium supplemented with 1000IU/mL of recombinant human IL-2 (Novartis Pharmaceuticals UK Ltd, Surrey, UK). Treg were stimulated with αCD3αCD28 T cell activator beads (ThermoFisher Scientific, Waltham,

Massachusetts, USA) on day 0 (at a ratio of three beads to one cell) and day 7 (at a ratio of one bead to one cell). Cultures were passaged and medium changed as required. Cells were rested without beads in medium supplemented with 200 U/mL of IL-2 for 48 h prior to assay.

For experiments intended for gene-editing, isolated Tregs were initially stimulated with MACS® GMP ExpAct™ Treg Kit at a bead:cell ratio of 4:1 (Miltenyi Biotech, Bergisch Gladbach, Germany) 1 day after isolation. Five days following isolation, cells were re-stimulated at a 1:1 bead:cell ratio. Beads were removed prior to non-viral gene editing using a strong magnet stand. Electroporated cells were rested overnight without beads before re-stimulation at a 1:1 bead:cell ratio. For further expansion rounds, Tregs were split and re-stimulated on alternate days at a 1:1 bead:cell ratio. All gene edited and unedited Tregs were cryopreserved on day 23 following cell isolation and were thawed as required.

### Generation of HLA-E knock-in construct

A dsDNA homology directed repair template (HDRT) was used for the targeted insertion of HLA-E into the *B2M* locus, creating a B2M-HLA-E fusion protein and disrupting the endogenous *B2M* gene. Homology arms mediating the insertion into B2M exon 2 flanked the HDRT. The construct contained the remaining part of B2M exon 2 as well as exon 3 followed by a flexible (G4S)4 linker and the HLA-E transgene (Suppl. Data 2). HDRTs were generated as previously described[62]. In brief, multiple fragment InFusion cloning was performed according to the manufacturer's protocol (Clontech, Takara) with purified PCR fragments (Kapa Hotstart HiFi Polymerase Readymix, Roche) and using synthesized DNA (gBlocks, IDT). In-Fusion cloning strategies were planned with SnapGene (Insightful Science; snapgene.com). Sequence validation of HDR-donor-template-containing plasmids was performed by Sanger Sequencing (LGC Genomics, Berlin). The B2M-HLA-E template was amplified from the plasmid by PCR using the primers (B2M_F: 5´ aagctcatttggccagagtgg 3´ and B2M_R: 5´ agctagaggaagccagtaggtaag 3´). PCR products were purified and concentrated using paramagnetic beads (AMPure XP, Beckman Colter Genomics). HDRT concentrations were quantified using the Qubit 4 fluorometer (Thermo Fisher Scientific) and a Qubit™ dsDNA BR-Assay-Kit according to the manufacturer's protocol and adjusted to 1 µg/mL in nuclease-free water.

### Gene editing to modulate HLA surface expression on primary human Treg

Gene editing in Treg was performed as previously described[40,41]. After 7 days of culture MACS® GMP ExpAct™ Treg Kit beads were depleted using a MACSiMAG™ Separator (Miltenyi). Cells were removed from the magnet, counted, and then washed twice in sterile PBS by centrifugation at $100 \times g$ for 10 min at room temperature (RT). In parallel, Cas9 RNP was prepared. For the electroporation of $10^6$ primary Tregs, 0.5 µL of poly(L-glutamic acid) (PGA) (molecular weight 15,000-50,000, Sigma-Aldrich, 100 µg/µL), 0.48 µL of synthetic modified sgRNA (Suppl. Data 3; 100 µM in TE buffer; IDT), and 0.4 µL recombinant SpCas9 protein (Alt-R S.p. Cas9 Nuclease V3; IDT; 61 µM) were mixed by thorough pipetting. The mixture was incubated for 15 min at RT and placed on ice. For HLA-E KI conditions, 0.5 µL of HDRT (stock concentration: 1 µg/µL) was added prior to electroporation. $10^6$ harvested Tregs were resuspended in 20 µL ice-cold P3 electroporation buffer (Lonza) just before electroporation to keep the exposure time to the electroporation buffers to a minimum. 20 µL of resuspended cells were transferred to the RNP/HDRT suspension, mixed thoroughly, and transferred into a 16-well electroporation strip (Lonza) without any air bubbles. The cells were electroporated using the EH-115 program on the 4D-Nucleofector (Lonza). Immediately after electroporation, 90 µL of pre-warmed Treg medium was added per well. After 10 min, the cells were carefully resuspended and transferred to two 96-well round-bottom plates (50 µL/well) containing 150 µL pre-warmed

Treg medium per well. The gene editing of HLA-E KI into *B2M* plus KO of *CIITA* Tregs (H/C Tregs) was performed in two subsequent editing steps. Seven days after HLA-E KI into *B2M* a second electroporation was performed using adenine base editor ABE8.20-m[30] mRNA produced in house by in vitro transcription as described previously[55]. Tregs were harvested, beads were magnetically removed, and cells were washed twice in PBS. $5 \times 10^6$ cells were resuspended in 100 µl P3 electroporation buffer (Lonza) and mixed with 2 µg of ABE8.20-m mRNA and 0.48 µL of synthetic modified sgRNA (100 µM in TE buffer; IDT). The suspension was electroporated in 100 µl Nucleocuvette Vessels (Lonza) using a Lonza 4D nucleofector device (program EH-115). 900 µL of pre-warmed Treg medium was added per cuvette. After 10 min, the cells were carefully resuspended and transferred to a 24 well cell culture plate.

### Flow cytometry

7-AAD viability staining solution (Affymetrix, Santa Clara, CA, U.S.), fixable blue dead cell stain kit (Thermo Fisher, U.S.), or Zombie NIR fixable viability kit (BioLegend) were used to eliminate dead cells from analysis. For phenotyping the following antibodies (supplier, clone) were used: mouse anti-human CD127 PE (BD, hIL-7R-M21), mouse anti-human CD25 PE-Cy7 (BD, M-A251), mouse anti-human CD3 eFluor450 (Affymetrix, OKT3), mouse anti-human CD4 ECD (Beckmann Colter, SFCI12T4D11), mouse anti-human CD45 APC (BD, RPA-T4), mouse anti-human CD45 APC (Affymetrix, H130), mouse anti-human CD8 FITC (Affymetrix, SK1), mouse anti-human CD8 PE (BD, HIT8a), mouse anti-human CD8 APC-Cy7 (BD, SK1), rat anti-human FOXP3 FITC (Affymetrix, PCH101), rat anti-human FOXP3 PE (Affymetrix, PCH101), rat anti-human FOXP3 eFluor450 (Affymetrix, PCH101), rat anti-mouse CD45 PE (Affymetrix, 30F11), mouse anti-human HLA-A,B,C PE-Cy7 (BioLegend, W6/32), mouse anti-human HLA-E APC (BioLegend, 3D12), mouse anti-human HLA-DR,DP,DQ FITC (BioLegend, Tü39), mouse anti-human FOXP3 FITC (BD Pharmingen, 259D/C7), mouse anti-human CD25 PC7 (Beckman, B1.49.9), mouse anti-human TNFα A700 (BioLegend, Mab11), mouse anti-human IFNγ APC-eF780 (Invitrogen, 4S.B3), rat IL-2 PE-Cy7 (BioLegend, MQ1-17H12), mouse anti-human CD4 PE (Beckman, 13B8.2), mouse anti-human CD8 PE-Cy7 (BD Pharmingen, RPA-T8), mouse anti-human CD3 PB (BioLegend, UCHT1), mouse anti-human CD56 A647 (BioLegend, 5.1H11), recombinant human anti-human NKG2A FITC (Miltenyi, REA110), recombinant human anti-human NKG2C PE (Miltenyi, REA205), mouse anti-human CD4 PE-eF610 (Invitrogen, RPA-T4), rat anti-mouse CD45 eF450 (Invitrogen 30-F11), mouse anti-human CD3 BV605 (BioLegend, OKT3), mouse anti-human HLA-DR BV650 (BioLegend, L243), mouse anti-human CD8 BV711 (BioLegend, SK1). Fluorescence was measured using BD FACSCanto, Beckman Colter CytoFLEX, or ThermoFisher Attune NxT flow cytometers.

### In vitro suppression assay

Responder PBMCs were stained with 10 µM CFSE (Thermo Fisher Scientific, Waltham, MA, USA) according to the manufacturer's protocol and cultured ($1 \times 10^5$ per well) for 72 h with autologous or allogeneic in vitro-expanded Treg, at ratios ranging from 1:1 to 1:8 Treg:responders. αCD3αCD28 T cell activator beads (Thermo Fisher Scientific) were added at a 1:5 bead:cell ratio. All assays were performed in triplicate. For analysis of responder proliferation from CFSE dilution profiles, division indices were calculated. The number of Treg (7AAD⁻CFSE⁻CD3⁺CD4⁺ cells) surviving following assay was expressed as a proportion of the initially plated cells. Treg enumeration was performed for 6 assays each using Treg from separate donors with 2−3 replicates per condition.

### Mice

BALB/cRag2⁻/⁻cγc⁻/⁻ mice (Jackson Laboratory, Bar Harbor, ME, USA) were housed in individually ventilated cages in the John Radcliffe

Hospital Biomedical Services Unit under specific pathogen-free conditions. All protocols were conducted in accordance with the UK Animals (Scientific Procedures) Act (1986) and approved by Oxford University's Committee on Animal Care and Ethical Review.

## Humanized mouse model of skin allograft rejection

Human skin was procured with full informed written consent and with ethical approval from the Oxfordshire Research Ethics Committee (REC B), study number 07/H0605/130. Surgical grafting of a 1 cm$^2$ human split-thickness skin graft onto the flank of BALB/cRag2$^{-/-}$cγc$^{-/-}$ female recipients was performed as previously described[26]. Five weeks following skin grafting, mice received $5 \times 10^6$ cryopreserved PBMC in pure RPMI via intra-peritoneal injection, with or without $1 \times 10^6$ or $5 \times 10^6$ in vitro-expanded Treg. Treg were either autologous or allogeneic to the PBMC donor. Both PBMCs and Treg were allogeneic to the skin graft. Grafts were observed for macroscopic markers of rejection by an assessor blinded to treatment group. To assess for long-term graft survival, mice were sacrificed following either graft rejection or at day 100 post-transplantation (whichever came sooner) by cervical dislocation. To evaluate tissue responses to adoptive transfer, mice were sacrificed by cervical dislocation at day 15. Blood was harvested from the inferior vena cava following schedule one for flow cytometric analysis. Grafts were dissected and fixed for 1 h in 10% neutral buffered formalin (CellStor). Treatments were allocated evenly between cages and litters. Raw data processing was performed by a researcher blinded to treatment group.

## In vivo mixed leukocyte assay

Treg were stained with 10 µM CFSE (Thermo Fisher Scientific). BALB/cRag2$^{-/-}$cγc$^{-/-}$ mice received $5 \times 10^6$ cryopreserved PBMCs in pure RPMI via intraperitoneal injection with or without $5 \times 10^6$ in vitro-expanded CFSE-labeled Treg. For each experiment, Treg were isolated from a single donor with allogeneic PBMC isolated from a second. Flow cytometric counting beads (Thermo Fisher Scientific) were co-injected with cells ($5 \times 10^5$ per mouse) to enable normalization of absolute cell counts between lavage samples. Seven days after injection of cells, peritoneal lavages were performed by flushing the peritoneal cavity with 10 mL saline following midline laparotomy and collecting the effluent. Cells extracted by lavage were analyzed by flow cytometry. For enumeration and phenotypic analyses, Treg were identified in lavage fluid by flow cytometry and CFSE staining. A total of 44 mice were used for mixed leukocyte assays, all of whom were female and aged between 8-12 weeks.

## HLA-matching

Peripheral blood underwent full tissue typing (HLA-A, -B, -C, -DR, and -DQ) by the Oxford Transplant Centre Histocompatibility and Genetics Laboratory.

## TSDR analysis

FOXP3-TSDR methylation analysis was performed by bisulfite amplicon sequencing as previously described[63]. Briefly, genomic DNA was isolated using the Quick-DNA Microprep Kit (Zymo Research D3020, Irvine, USA) according to the manufacturer´s protocol. Up to 200 ng genomic DNA was bisulfite-converted using EZ-DNA methylation Gold kit (Zymo Research D5005, Irvine, USA). Subsequently, PCRs were performed (10 µl of bisulfite-treated DNA, 2xKAPA HiFi Hotstart Uracil+ ReadyMix (Kapa Biosystems, USA, KK2802), 0.25 mM of each dNTP, using 0.3 pmol of primers (F1: 5′ ACACTCTTTCCCTACACGACGCTCTTCCGAT CTTTTGGGGGTAGAGGATTTAGAGGG-3′ and R3: 5′GACTGGAGTTCA GACGTGTGCTCTTCCGATCTCCACATCCACCAACACCCAT -3′). Amplicons were purified with QIAquick PCR Purification Kit (Qiagen Germany, 28106), normalized to 20 ng/µl, and sequenced ($2 \times 300$ bp paired-end). Reads were aligned and evaluated using the Bismark package[64].

## Generation of alloreactive T cell lines

PBMCs were isolated from three healthy donors. CD56$^+$ cells were depleted from whole PBMCs using CD56 microbeads (Miltenyi) according to the manufacturer's protocol. As feeder cells, CD3-depleted PBMCs from two Treg donors were irradiated (30 Gy). Feeder cells were cultured with CD56-depleted PBMCs at a ratio of 1:1 for 9 days in T cell medium (RPMI1640 supplemented with 10% FCS, 10 ng/mL IL-7, and 5 ng/mL IL-15). CD3 enrichment was performed on day 9 prior to a second round of stimulation with irradiated feeder cells at a ratio of 1:1 until day 20. T cells were split every other day. Lineage phenotyping was performed using αCD3 Pacific Blue, αCD56 AF647, αβTCR PE, γδTCR FITC, αCD4 APC-Fire750, and αCD8 BV510 (see above for details). Alloreactive T cells were cultured in cytokine-free medium overnight prior to the cytotoxic assay.

## In vitro alloreactive T cell cytotoxic assay

Expanded alloreactive T cells were co-cultured with wild type or gene-edited allogeneic Tregs. Alloreactive T (alloT) cells were stained with CFSE (Thermo Fisher Scientific) prior to co-culture. AlloT cells were added to 25,000 target T cells in 96-well, round-bottom cell culture plates at various alloT:Treg ratios (2:1, 4:1, 8:1) with control wells containing only the target cells. Plates were centrifuged at $100 \times g$ for 1 min at RT and incubated at 37 °C and 5% CO$_2$. A 96 well cell culture plate containing 50 µL of PBS with DAPI (final dilution 1:100,000, stock concentration: 1 mg/mL, Thermo Scientific) was prepared and cooled to 4 °C for 30 min prior to stopping the assay. 22 h after setting up the co-culture, 50 µL of resuspended cells were added to the 50 µL of PBS/DAPI solution, followed by a minimum of 10 min incubation at 4 °C. 30 µL of the cell suspension was analyzed by flow cytometry to evaluate the number of viable cells. Alloreactive T cell-mediated cytotoxicity was calculated as the reduction in cell number of CFSE-negative target cells in the co-culture normalized to the target only control.

## In vitro primary NK cell cytolytic assay

NK cells were freshly isolated from the PBMCs of four healthy donors using the NK isolation kit (Miltenyi, Germany) according to the manufacturer's instructions. Isolated NK cells were cultured in NK medium and were expanded prior to phenotypic characterization by flow cytometry (see above for info). Six days following isolation, expanded NK cells were used in the in vitro cytotoxicity assay. In brief, 100,000 target cells (gene edited Treg, unedited Treg, or K562 controls) were co-cultured at various NK:target cell ratios (0.25:1, 0.5:1, 1:1) for 20 h before flow cytometry. Of the 200 µL cell suspension, 50 µL was used to establish the number of viable target cells, whilst 150 µL was used to establish target HLA-A,B,C (PE-Cy7) and HLA-E (APC) expression.

## Spatial transcriptomics and immunohistochemistry

FFPE sample blocks were sectioned at 5 µm onto Xenium slides (10X Genomics) that had been equilibrated to RT for 30 min under RNAse free conditions. Sections were baked at 42 °C for 3 h then stored in a desiccator overnight until processing. The Xenium assay was performed according to the manufacturer's recommended protocol (CG000580). Briefly, slides were cleared, rehydrated, and incubated overnight with the 377 gene Human Multi-Tissue and Cancer Xenium Pre-Designed Gene Expression Panel (10x Genomics, 1000626) and 50-gene custom-designed probe set. Slides subsequently underwent rolling circle amplification, chemical autofluorescence quenching, and DAPI and cell segmentation staining prior to imaging on the Xenium Analyzer. Count matrices were merged in Seurat (version 5.1.0), and quality control performed to remove cells with no unique features and genes with zero expression[65]. Counts were normalized to 10,000 counts per cell and then natural log ($\log_{1p}$) transformed. Label transfer from reference single cell RNA sequencing data was performed after identification of transfer anchors[66]. Differential gene expression was performed with DESeq2 (version 1.46.0) on pseudobulks of grafts

stratified by damage and visualized with pheatmap (version 1.0.12)[67]. Visualization was performed with SpatialData and NapariViewer with co-ordinate harmonization for orientation[68].

Sequential 5 μm sections were taken to standard SuperFrost histology slides and were dried overnight at 37 °C before baking for 4 h at 60 °C. Paraffin was cleared with xylene washes before slide rehydration across an ethanol gradient. Heat-induced epitope retrieval was performed at pH 9 with Tris-EDTA for 20 min. Tissues were blocked with 10% normal goat serum for 1 h at room temperature, then 3% hydrogen peroxide for 15 min, before incubation with anti-human CD8 (clone: CAL66) at 1:500 dilution overnight at 4 °C. Goat anti-rabbit IgG secondary antibodies were incubated for 1 h at room temperature and incubated with 3, 3'-diaminobenzidine for 5 min before washing and hematoxylin counterstaining for 15 s. Slides were scanned using a Zeiss AxioScan7, and positive cells were quantified with QuPath (version 0.5.1).

## Statistics and reproducibility

All data were analyzed, and graphs produced, using GraphPad Prism version 9.0 or version 10.3.1 for MacOS (GraphPad Software, La Jolla, California, USA). For in vitro Treg suppression assays, repeated measures two-way ANOVAs were used to assess statistical significance, with the Bonferroni *post hoc* correction applied for pairwise comparisons between treatments. Student's *T*-test or Mann–Whitney U test was applied to comparisons of cell frequencies between autologous and allogenic PBMC/Treg combinations. Graft survival kinetics were assessed using Mantel–Cox log-rank tests for each pairwise comparison between groups. All plotted error bars represent mean ± standard error of the mean (SEM) unless otherwise stated. No statistical method was used to predetermine sample size, and no data were excluded from analyses. Experimental conditions were randomized across cages and litters, and the investigators were blinded during outcome assessment. Investigators were not blinded to experimental group for in vitro assays.

## Reporting summary

Further information on research design is available in the Nature Portfolio Reporting Summary linked to this article.

## Data availability

All data are presented in the main text or the supplementary materials. The spatial transcriptomic datasets (Fig. 5 and Supplementary Fig. 5) generated in this study have been deposited in the FigShare database (https://doi.org/10.6084/m9.figshare.28988606). Source data are provided with this paper.

## Code availability

Analyses in this study used pre-existing tools and did not generate original code.

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

## Acknowledgements

European Union, Horizon 2020, grant agreement: 825392 (FI, JH, OM, WD, VG, JKP, MSH, HDV, PR, DLW). ERC starting grant 'EpiTune', grant agreement 803992 (JKP). Medical Research Council, Clinical Research Training Fellowship, award reference: MR/V000942/1 (OM). Wellcome Trust 211122/Z/18 (FI). Kidney Research UK SF1/2014 (JH). British Heart Foundation, FS/ 12/72/29754 (KM). Berlin Institute of Health, SPARK-BIH (JK, DLW). Oxford Transplant Foundation Research Grant (OM, SS). Chinese Academy of Medical Sciences (CAMS) Innovation Fund for Medical Science (CIFMS), China. Grant number: 2024-I2M-2-001-1 (FI).

## Author contributions

Conceptualization: O.M., W.D., V.G., J.H., F.I., and D.L.W. Data curation: O.M., K.M., W.D., V.G., S.S., M.B., A.C., H.S., C.F., M.Y., J.K.P., J.H., F.I., and D.L.W. Formal Analysis: O.M., W.D., V.G., S.S., C.F., and M.Y. Funding acquisition: J.H., F.I., J.K.P., H.D.V., P.R., and D.L.W. Investigation: O.M., K.M., W.D., V.G., C.F., J.K., M.V., and M.Y. Methodology: O.M., K.M., W.D., V.G., C.F., J.K., M.V., A.K., L.A., J.K.P., M.S.H., P.R., H.D.V., and D.L.W. Project administration: J.H., F.I., M.S.H., H.D.V., P.R., and D.L.W. Resources: A.K., J.K.P., L.A., M.S.H., P.R., H.D.V., J.H., F.I., and D.L.W. Software: not applicable. Supervision: J.H., F.I., and D.L.W. Validation: O.M., W.D., V.G., C.F., J.K., and M.V. Visualization: O.M., W.D., V.G., S.S., J.H., F.I., and D.L.W. Writing—original draft: O.M., W.D., V.G., H.D.V., J.H., F.I., and D.L.W. Writing —review and editing: O.M., W.D., V.G., HDV, J.K.P., J.H., F.I., and D.L.W.

## Competing interests

HDV is the founder and C.S.O. of CheckImmune GmbH. H.D.V., P.R., and D.L.W. are co-founders of TCBalance Biopharmaceutical GmbH. All other authors declare that they have no other competing interests.

## Additional information

[1]Translational Research and Immunology Group, Nuffield Department of Surgical Sciences, University of Oxford, Oxford, UK. [2]Berlin Center for Advanced Therapies (BeCAT), Charité – Universitätsmedizin Berlin, corporate member of Freie Universität Berlin and Humboldt-Universität zu Berlin, Berlin, Germany. [3]BIH-Center for Regenerative Therapies (BCRT), Berlin Institute of Health at Charité – Universitätsmedizin Berlin, Berlin, Germany. [4]Institute of Medical Immunology, Charité – Universitätsmedizin Berlin, corporate member of Freie Universität Berlin and Humboldt-Universität zu Berlin, Berlin, Germany. [5]Center for Cell and Gene Therapy, Baylor College of Medicine, Houston, TX, USA. [6]Department of Pediatric Hematology/Oncology, Charité – Universitätsmedizin Berlin, corporate member of Freie Universität Berlin and Humboldt-Universität zu Berlin, Berlin, Germany. [7]German Cancer Consortium (DKTK), Berlin, Germany. [8]Department of Molecular and Cellular Biology, Baylor College of Medicine, Houston, TX, USA. [9]Dan L Duncan Comprehensive Cancer Center, Baylor College of Medicine, Houston, TX, USA. [10]These authors contributed equally: Oliver McCallion, Weijie Du, Viktor Glaser. [11]These authors jointly supervised this work: Dimitrios L. Wagner, Joanna Hester, Fadi Issa. ✉e-mail: dimitrios-l.wagner@charite.de; joanna.hester@nds.ox.ac.uk; fadi.issa@nds.ox.ac.uk

