## [Transparent Peer Review file · Nature Communications]

HLA matching or genetic engineering enables effective allogeneic human regulatory T cell therapy

Corresponding Author: Professor Fadi Issa

Version 0:

Reviewer comments:

Reviewer #1

(Remarks to the Author)
August 29, 2023

To the Editor:

In their manuscript entitled "Matching or genetic engineering of HLA Class I and II facilitates successful allogeneic 'off-the-shelf' regulatory T cell therapy", authors McCallion, Issa and colleagues describe a regulatory T cell (Treg) engineering platform that could markedly enhance the therapeutic use of this suppressive cell type in diverse organ and stem cell transplant settings as well as in autoimmune disease. Their finding is very pertinent because, while autologous, adoptively transferred Treg were shown to persist long-term in patients with diabetes (Bluestone SciTranMed '15), persistence after adoptive transfer of allogeneic Treg was much shorter (Parmar Oncotarget '18). The authors optimized allogeneic Treg therapy through HLA engineering, which produced Treg with the same efficacy as autologous Treg.

This is a timely and consequential manuscript, and the rigor of their Treg expansion and gene manipulation was more than acceptable. There is one major point and a few minor points that need to be addressed.

Major point:

- Figure 5 defines the whole paper. While I understand that the Log-Rank test for trend is $p=0.04$, neither of the more traditional measures are significant. This appears to be an $n=1$, with only 5 mice per group. I think it really needs to be repeated, and hopefully the frozen/thawed banking approach will allow this to be done without too much trouble.

Minor points:

- The authors have documented an inverse correlation between CD70 expression and suppressive function in a previous manuscript, could the authors please comment on CD70 expression relative to the current manuscript.
- Similarly, the authors have previously shown Treg decrease PBMC-derived CD4 and CD8 division index in their xeno IP model. In the current study, could the authors comment on whether allogeneic Treg able to suppress PBMC-derived T cell division equivalent to autologous, or did CD8-mediated depletion inhibit Treg suppression?
- The Treg suppression demonstrated in Figure 4I (and the Foxp3 expression shown in 4F) demonstrate the quality of the cultures. However, to demonstrate equivalent suppressive function amongst the 6 groups, it would be better to show a series of dilutions that more readily define the suppression curve as in Fig 1 A and B.
- Figure 5 (cont.): It would also be good to report the specifics on Foxp3 and in vitro suppressive function for the frozen/thawed cultures that are used in vivo. Finally, is there a measure that could be followed that would be predictive for disease severity. For example, PBMC-derived T cell expansion in peripheral blood is monitored in many xenoGVHD models. If human cells are present in PB with your IP model, it would provide an opportunity to interrogate the persistence/purity of the alloH/C Treg.

Reviewer #2

(Remarks to the Author)

Using Tregs as a cell-based therapy in transplantation remains a promising strategy. Great efforts are being made to

develop superior Treg products, including engineered cells with altered specificities, enhanced stability or suppression, favorable trafficking, or better engraftment and survival after administration. This work contributes to these efforts by developing Tregs for allogeneic applications that lack MHC Class I and II expression to avoid allogeneic rejection, or express an HLA-E-B2M transgenic receptor to evade NK cells. These are neat technological ideas, which the authors imagine using in off-the-shelf allogeneic Treg cell therapy products. By their nature, the experiments are complicated and I wasn't able to properly understand their design from the descriptions. It is a concern that general conclusions were seemingly drawn from small and potentially unrepresentative samples. Unfortunately, the final results in Fig.5, do not prove an advantage of engineered Tregs over fully mismatched allogeneic Tregs, which negates the main conclusions.

Results: The suppressive capacity of autologous and allogeneic Tregs were first compared using an in vitro T cell proliferation assay. Their suppressive capacity was then tested using a long-established human skin transplant model in immunodeficient mice. This led to the claim that autologous Tregs are more effective than allogeneic Tregs in preventing allograft rejection, whereas they are equally suppressive in culture.

1. The design of experiments shown in Fig.1A-D is not explicit. The reason for sample pairing is unclear. Are the six allogeneic responders actually independent samples, or are they 2 or 3 pairs of responders in combination with 2 or 3 stimulators? The small sample size and possible dependence are concerns.
2. In Fig.1A-D and throughout, were responder-stimulator pairs HLA-typed – even crudely – and was this information used to select HLA-DR mismatched pairs?
3. In Fig.S1, the frequency and number of cells in the CFSE+ and CFSE- regions are not given. Are these consistent with results in Fig.2A?
4. The human-to-mouse skin transplant model shown in Fig.3E was established by the Oxford group and is well-published. However, in the context of this manuscript about HLA-typing, its interpretation is complicated. As I understand the model, in the allogeneic Treg setting, there are 3 allogeneic donors – of skin, PBMC and Tregs. It is a concern that partial matching between PBMC or Tregs with skin is an uncontrolled confounder in this work.
5. The design of experiments shown in Fig.1F-G is also not explicit. 30 mice were investigated. (a) did all 30 mice receive skin transplants from the same donor? (b) did all mice receive PBMC from the same donor? (c) were skin and PBMC fully or partially HLA-matched? (d) were all allogeneic Tregs from the same PBMC donor? (e) are there any statistically dependent donor-recipient combinations in this experiment? (f) why were 30 mice not equally distributed into 3 groups? Does this hint at unexpected low Treg yields after expansion? It would be a concern if this experiment represented only n=1 allogeneic and n=1 autologous Treg donor because the observed differences in graft survival might reflect the quality of Treg preparations, or some other donor-related factor, rather than a difference in allogeneicity.

Results: To explain why autologous Tregs are more effective in vivo than allogeneic Tregs, the survival of Tregs was investigated in allogeneic PBMC cocultures and a reconstitution model in mice. Depletion of CD8+ T cells from PBMC led to increased survival of allogeneic Tregs in vivo.

6. Does Fig.2A show a further analysis of the same experiment shown in Fig.1-D? The design of this experiment is not explicit. Are these independent combinations, or just permuted pairs of donors and recipients? The variability between samples is extreme (range ~20-80% surviving Treg) so differences between autologous and allogeneic Tregs may be impossible to detect given n=6. It is a concern that Treg survival in these experiments might represent an unidentified difference in the quality of Treg preparations.
7. Tregs were only enumerated in peritoneal wash-outs. Is this representative of all other locations in recipient mice? It is a concern that allogeneic Tregs might not have been retained in the peritoneal cavity in the same way as autologous Tregs.
8. The design of experiment shown in Fig.2C is not explicit. Are all autologous Treg samples from n=1 donor? Are all allogeneic Treg samples from n=1 donor?
9. The design of experiments shown in Fig.2D-F is not explicit. Are all autologous Treg samples from n=1 donor? Are all allogeneic Treg samples from n=1 donor?
10. In Fig.2F, does NK cell depletion actually increase % undivided Tregs, but the sample size was too small to detect a difference? I am concerned that the conclusion on L128 is not supported.
11. The depletion experiments in Fig.2D-F shows that CD8+ T cells are necessary for low Treg numbers in peritoneal wash-outs, but does not show they are sufficient. The conclusion on L129 that CD8+ T cells are predominantly responsible for the effect is not fully supported.
12. It is not explained why CD8+ T cells kill allogeneic Tregs in vivo but not in vitro.
13. From a translational perspective, Tregs are unlikely to be administered to an otherwise untreated transplant recipient. What is the effect of low-dose tacrolimus in this system? Does it restore the suppressive capacity of allogeneic Tregs in the transplant model?

Results: To isolate HLA-matching as an explanation for the relative effectiveness of autologous and allogeneic Tregs, 3 pairs of PBMC-Treg donor pairs were identified with varying degrees of Class I and II matching. These pairs were used in the skin transplant model.

14. The design of the experiment shown in Fig.2A-C is not explicit. Which of the donors was the Treg donor and which was the PBMC donor? This is a concern because D209 is present in two pairs.
15. Minor: Row titles in Table S1 should be checked.
16. On L131, it is reported that ">150 Treg/PBMC pairs" were screened. This is unclear. There are 153 possible pairings of just 18 donors. Were these donors from the Transfusion Service or lab volunteers?
17. On L136, the number of PBMC was reduced for Fig.3B compared to Fig.1F to "create a more challenging model." Why was this necessary and how does this statement affect the interpretation of Fig.1F?

18. Differences in allograft survival times are reported, which are attributed to differences in HLA-matching between PBMC and Treg. However, given the small number of PBMC donors (n=2 or 3), this might also reflect differences in the allogeneicity of the transplanted tissue. It is a concern that the conclusion stated in the title of Fig.3 (L879) is not supported by the results.

19. In Fig.3B, censoring of 2/5 animals in the “partial mismatch” group is problematic. We cannot know these censored recipients would’ve rejected within 100 days. More data is required to support the stated conclusions.

20. Xeno-GVHD was not reported in experiments shown in the untreated controls or allogeneic Treg recipients in Fig.1F. The post hoc conclusion stated on L142-143 is speculative given the small sample size of n=1 cell preparation.

Results: Tregs were genetically manipulated to lack MHC Class I, II or I+II, and/or express an HLA-E-B2M fusion protein. The phenotype and functionality of these Tregs was tested in vitro. Evidence of genetic manipulation in Fig.4B is convincing, as are phenotypes in Fig.4D-G and J.

21. It is unfortunate that conditions for the suppression assay were changed from Fig.1A-B (3 days; 1:5 bead:cell ratio) to Fig.4I (5 days; 1:1 bead ratio). Nevertheless, unmanipulated autologous and allogeneic Treg controls are shown in Fig.4I, so the conclusion of preserved suppressor function stands.

22. The stability of modified Treg populations over long culture periods or following in vivo transfer was not studied. Without alloimmune selection pressure, are these cells stable in vitro and in vivo?

Results: To test whether silencing MHC Class I expression increased Treg susceptibility to NK cell-mediated cytotoxicity, primary human NK cells were expanded and used in killing assays. Results shown in Fig.S3 are complete and convincing.

23. Minor: In Fig.S3E-F, consider using the same scale on y-axis to emphasize the differences of interest.

Results: To test whether silencing MHC Class I+II expression, or forced expression of HLA-E, increased Treg susceptibility to NK cell-mediated cytotoxicity, alloreactive polyclonal T cells were generated over a 20-day expansion period against irradiated donor PBMC. Then an overnight killing assay was performed against Tregs from the same donors used for MLR. Results shown in Fig.S4A-G are technical controls and are convincing. Silencing MHC I+II reduced cytotoxic activity against allogeneic Tregs. Expressing HLA-E had no effect.

24. Cytotoxicity is reduced, but not prevented, by silencing MHC I+II. There are several possible explanations, including MHC-unrestricted killing or incomplete silencing. An autologous Treg control might help to assess LAK-like unspecific killing. Is it perhaps a limitation of the whole approach that H/C Tregs were not superior to DKO Tregs in this system?

25. Minor: It took some moments to understand the unit, “% protection from cytolysis normalized to WT Tregs,” shown in Fig.S4I. Is this not the same as the ratio of modified Treg numbers to unmodified Treg numbers, expressed as %?

Results: In a final experiment, the efficacy of autologous Tregs, unmodified allogeneic Tregs and modified allogeneic Tregs were compared using the skin transplant model.

26. On L198, a claim is made that conditions for the model were selected to, “mimic a true off-the-shelf scenario.” However, it stated on L196 that PBMC and Treg donors were fully mismatched. This is rarely the case in modern transplantation. This set-up is, in fact, an extreme constellation that should favor MHC I+II-deficient Tregs over a partially matched donor-recipient pairing – a conclusion drawn by the authors themselves from results in Fig.3.

27. The group sizes reported in the figure legend and survival curves are not concordant. Specifically, the plot shows n=6 mice, whereas the legend reports n=5. This is a strange error to make, which raises a major concern about selective reporting of results. The statement on L201 that 40% recipients of H/C Tregs survived to d100 is not supported by Fig.5B.

28. The legend for Fig.5 refers only to panels (A) and (F). I have never seen this manuscript before, but I assume panels were removed from an earlier version. See (27).

29. The group sizes are likely too small to reach a statistical conclusion; however, it seems that autologous Tregs worked better than alloH/C Tregs in preventing allograft rejection. Thinking translationally, these results aren’t overwhelmingly evidence for DKO or H/C Tregs as an off-the-shelf alternative to autologous Tregs.

Discussion: This is nicely written, but feels more like a review than discussion. Care must be taken not to misrepresent the results of the study.

30. On L234, a claim is made that, “we demonstrate that [modified] Tregs survive for long enough to establish a regulatory environment.” This article reports 7 day survival of unmodified Tregs in the peritoneal cavity. It does not examine the location or duration of Tregs at any site. It does report on survival of genetically modified Tregs at all.

31. On L237, a claim is made that, “this provides the first evidence that the therapeutic efficacy of allogeneic Tregs can be significantly enhanced by matching and/or gene editing, paving the way for an improved ‘off-the-shelf’ strategy to mass manufacture an allogeneic Treg cellular therapy.” The results in Fig.5 do not show a significant improvement in efficacy of modified Tregs. There is no proof of an improved manufacturing strategy.

32. On L250, the discussion about antigen-specific, TCR specificity-dependent activation of Tregs and their non-specific, TCR-independent suppression of T cells in vitro seems muddled.

33. On L274, the need for a “genetic safety switch” is mentioned owing to modified Tregs’ severely compromised antigen presentation. What risks are envisaged? Is there a special concern about cell product-derived malignancies? The authors should be more explicit about the perceived risks and road-blocks to this technology.

34. On L277-287, gene editing in iPSC-derived Tregs was not addressed in the results and should be omitted from the discussion.

35. On L289, if the possibility of allogeneic Treg MHC-matching is going to be raised, then the authors must seriously discuss the logistics and economics of a matching program. The complexity of HSC transplantation shows that this is not trivial. In my view, this is a completely impractical strategy.

36. On L292-294, an untrue statement is made about the results. "Our results demonstrate that both of these [matching and gene editing] approaches facilitate the long-term survival of Tregs in vivo resulting in a therapeutic effect." Nowhere in this manuscript is it shown that transferred Tregs survive for any longer than 7 days. This observation is restricted to unmodified autologous Tregs, not matched allogeneic Tregs or gene-edited Tregs.

37. On L297-299, it is stated that, "our findings ... also open the door to realizing the full potential of this transformative therapeutic approach for a variety of immune-mediated pathologies." This study does not examine pathological immune responses apart from allogeneic reactions, either in vitro or in vivo. The conclusion is speculative and empty.

Reviewer #3

(Remarks to the Author)

McCallion and colleagues evaluate allogeneic regulatory T cells (Tregs) as potential 'off the shelf' cell product to suppress allo-immune responses. In lymphopenic mice (BALB/c Rag2^{-/-} cγc^{-/-}) that were reconstituted with human PBMC, allogeneic Tregs (i.e. derived from another source than the PBMC) were less potent than autologous Tregs (i.e. derived from the PBMC) in prolonging human skin graft survival. Reduced efficacy was associated with lower in vivo survival of allogeneic Tregs due to CD8 T cell- and to a lesser degree NK cell-dependent elimination. To overcome this limitation the authors test two approaches: HLA matching and genetic engineering. Both (incomplete) HLA-matching and silencing of HLA class I and II expression combined with the transgenic expression of HLA-E (to modulate NK reactivity) partially restored the efficacy of allogeneic Tregs in the humanized skin allograft model. The genetically engineered cell product developed by the investigators is novel and of interest as it represents a relevant step towards the development of 'off the shelf Tregs'. Several issues remain:

1. The humanized mouse model used in this report appears volatile: In figure 5B, the estimated MST for the 'autoTreg' group (serving as positive control) is approximately 55 days, while in figure 3B 'partially HLA-matched' allogeneic Tregs demonstrate an MST >100 days (60% long-term survival) at the same Treg:PBMC ratio. Is there any methodological explanation for the observed variability of graft survival in this model? This variability makes robust conclusions difficult in the context of the low sample sizes tested in the current study.

2. The difference in skin graft survival between the groups receiving unmanipulated allogeneic Tregs ('PBMC + alloTreg') and the genetically engineered allogeneic Tregs ('PBMC + alloH/C Treg') is modest. Given the high variability of graft survival in this model (as described above), a larger number of animals per group and a more complete statistical testing comparing all relevant therapeutic groups (including allo vs. alloH/C Tregs), would be needed for a more robust conclusion. To sustain the conclusion that H/C Tregs are comparable to autoTregs (lines 200-202) additional experiments would be necessary.

3. Substantial expression of HLA class I and II antigens persists in gene-edited allogeneic Tregs, both in DKO and in H/C cells (Fig 4 C+D). This relevant limitation needs to be discussed.

4. Assessing potential allo-sensitization against the 'Treg donor' would be important, especially since some expression of class I and class II HLA is retained in engineered Tregs.

5. The in vivo mixed lymphocyte reaction (intraperitoneal injection of PBMC together with allogeneic or autologous Tregs) is interesting, yet it does not allow to draw a definitive conclusion regarding the in vivo persistence of the injected Tregs under physiological conditions as the number of Tregs isolated via peritoneal lavage is the only readout. A more physiological approach, as for example injecting marked Tregs (together with PBMC) intravenously and tracking them throughout lymphatic tissues and particularly within the allograft would be needed to adequately investigate the in vivo persistence of the cell product.

6. Minor: Errors in the description of the panels in the legend of Fig. S3 need correction.

7. Minor: The depicted graft survival curve for the group 'PBMC + alloH/C Treg' in figure 5B (indicating 6 animals per group) does not match with the number of animals (n=5) mentioned in the figure legend and with the 40% long-term graft survival rate mentioned in the text.

Version 1:

Reviewer comments:

Reviewer #1

(Remarks to the Author)

Thank you for addressing the comments of all the reviewers.

Reviewer #3

(Remarks to the Author)

In the revised manuscript, the authors have addressed the limitations of their approach related to potential allo-sensitization and have incorporated additional experimental data to respond to concerns regarding the consistency of their humanized mouse model. The manuscript has been further strengthened, the findings are of high interest.

As pointed out during the initial review, we would have preferred if the authors had provided a formal statistical comparison of the skin graft survival between unmanipulated allogeneic Tregs (PBMC + allo Tregs) and the engineered allogeneic Treg product (PBMC + alloHC Treg) presented in Figure 5b – as this is the critical group comparison to claim efficacy of the

investigated engineering approach.

The authors now conclude that alloH/C Tregs are comparable to autoTregs (Fig 5), which more accurately reflects their data than stating equivalent outcome (as for instance in the abstract). The new wording should be used throughout the manuscript.

In new suppl. Fig 5A Tregs negative for HLA were followed. Since only the alloH/C Treg group was negative for HLA from the beginning, not much can be concluded with regard to comparing persistence between groups. Thus, the data as presented in their current form seem of limited value. Why not compare the persistence of Tregs across all groups with staining capturing all the different transferred Treg populations?

Version 2:

Reviewer comments:

Reviewer #3

(Remarks to the Author)

The authors have adequately addressed the remaining questions and resolved all pending issues. The additional plot now shown as Suppl. Fig. 5B increases the clarity of the manuscript, as does the refined wording.

We thank the Editor for the opportunity to revise our work and the three reviewers for their detailed and constructive feedback. We have addressed every point below and indicated where changes have been made in the manuscript (all edits are highlighted).

REVIEWER ONE

In their manuscript entitled “Matching or genetic engineering of HLA Class I and II facilitates successful allogeneic 'off-the-shelf' regulatory T cell therapy”, authors McCallion, Issa and colleagues describe a regulatory T cell (Treg) engineering platform that could markedly enhance the therapeutic use of this suppressive cell type in diverse organ and stem cell transplant settings as well as in autoimmune disease. Their finding is very pertinent because, while autologous, adoptively transferred Treg were shown to persist long-term in patients with diabetes (Bluestone SciTranMed '15), persistence after adoptive transfer of allogeneic Treg was much shorter (Parmar Oncotarget '18). The authors optimized allogeneic Treg therapy through HLA engineering, which produced Treg with the same efficacy as autologous Treg.

This is a timely and consequential manuscript, and the rigor of their Treg expansion and gene manipulation was more than acceptable. There is one major point and a few minor points that need to be addressed.

Major point:

1. Figure 5 defines the whole paper. While I understand that the Log-Rank test for trend is $p=0.04$, neither of the more traditional measures are significant. This appears to be an $n=1$, with only 5 mice per group. I think it really needs to be repeated, and hopefully the frozen/thawed banking approach will allow this to be done without too much trouble.

We thank the reviewer for underscoring the central importance of these *in vivo* data. We have now performed additional *in vivo* studies to confirm the results. In the revised manuscript, Figure 5 (panels C-G and Suppl. Figure 5) now includes repeated skin transplant studies using orthogonal methodologies: immunohistochemistry, spatial transcriptomics, and macroscopic graft assessment. The results demonstrate improved graft protection and correlate with reduced inflammatory signatures and immune cell infiltration in animals treated with HLA-engineered Tregs. In addition, peripheral blood analysis at day 21 (Suppl. Figure 5A-B) confirms the persistence of HLA-edited Tregs.

Minor points:

1. The authors have documented an inverse correlation between CD70 expression and suppressive function in a previous manuscript, could the authors please comment on CD70 expression relative to the current manuscript.

We thank the reviewer for raising this point. We now include CD70 data (Suppl. Fig. 4J) and note in the text that expression is equivalent across all Treg products.

2. Similarly, the authors have previously shown Treg decrease PBMC-derived CD4 and CD8 division index in their xeno IP model. In the current study, could the authors comment on whether allogeneic Treg able to suppress PBMC-derived T cell division equivalent to autologous, or did CD8-mediated depletion inhibit Treg suppression?

In Figure 2E-F we now clarify that CFSE-labelled Tregs were tracked post-injection, and CD8 depletion results in both higher recovery and a greater proportion of divided Tregs. As PBMCs were not labelled, we are unable to quantify their proliferation directly. We have clarified this limitation in the text.

3. The Treg suppression demonstrated in Figure 4I (and the Foxp3 expression shown in 4F) demonstrate the quality of the cultures. However, to demonstrate equivalent suppressive function amongst the 6 groups, it would be better to show a series of dilutions that more readily define the suppression curve as in Fig 1 A and B.

Thank you. We confirm that the experiments were performed using the same serial dilutions as in Figure 1 (1:2, 1:4, 1:8).

- Figure 5 (cont.): It would also be good to report the specifics on Foxp3 and in vitro suppressive function for the frozen/thawed cultures that are used in vivo.

We have updated the text to indicate that the in vitro proliferation, suppression, and phenotypic data we present in Figure 4 include the donor/recipient pair used in the skin transplant experiments (Figure 5).

- Finally, is there a measure that could be followed that would be predictive for disease severity. For example, PBMC-derived T cell expansion in peripheral blood is monitored in many xenoGVHD models. If human cells are present in PB with your IP model, it would provide an opportunity to interrogate the persistence/purity of the alloH/C Treg.

In response to the suggestion, we have now included flow cytometry data in Suppl. Figure 5A-B that demonstrate the persistence of allogeneic HLA-edited Tregs in peripheral blood (detectable as HLA-A, -B, -C-negative CD4+ cells), underscoring the potential to monitor persistence.

REVIEWER TWO

Using Tregs as a cell-based therapy in transplantation remains a promising strategy. Great efforts are being made to develop superior Treg products, including engineered cells with altered specificities, enhanced stability or suppression, favorable trafficking, or better engraftment and survival after administration. This work contributes to these efforts by developing Tregs for allogeneic applications that lack MHC Class I and II expression to avoid allogeneic rejection, or express an HLA-E-B2M transgenic receptor to evade NK cells. These are neat technological ideas, which the authors imagine using in off-the-shelf allogeneic Treg cell therapy products. By their nature, the experiments are complicated and I wasn't able to properly understand their design from the descriptions. It is a concern that general conclusions were seemingly drawn from small and potentially unrepresentative samples. Unfortunately, the final results in Fig.5, do not prove an advantage of engineered Tregs over fully mismatched allogeneic Tregs, which negates the main conclusions.

Results: The suppressive capacity of autologous and allogeneic Tregs were first compared using an in vitro T cell proliferation assay. Their suppressive capacity was then tested using a long-established human skin transplant model in immunodeficient mice. This led to the claim that autologous Tregs are more effective than allogeneic Tregs in preventing allograft rejection, whereas they are equally suppressive in culture.

- The design of experiments shown in Fig.1A-D is not explicit. The reason for sample pairing is unclear. Are the six allogeneic responders actually independent samples, or are they 2 or 3 pairs of responders in combination with 2 or 3 stimulators? The small sample size and possible dependence are concerns.

We have clarified in the figure legend that the responder-stimulator combinations were generated by permuting cells from three donors and that all pairings were HLA-typed (see below).

- In Fig.1A-D and throughout, were responder-stimulator pairs HLA-typed – even crudely – and was this information used to select HLA-DR mismatched pairs?

Yes, responder-stimulator pairs in this experiment were HLA-typed at the HLA-A, HLA-B, and HLA-DR loci which are known to be critical in alloresponse generation and were intentionally selected to be fully mismatched as shown below:

	A		B		DR	
Donor 1	11	1	38	60	1	13
Donor 2	3	2	18	7	15	-
Donor 3	24	-	7	-	7	11

We have updated the legend of Figure 1 to reflect this important detail and have included these data in Suppl. Fig. 1.

3. In Fig.S1, the frequency and number of cells in the CFSE+ and CFSE- regions are not given. Are these consistent with results in Fig.2A?

We have extended Suppl. Figure 1 (C-F) with quantitative data on the frequencies and absolute numbers of CFSE+ and CFSE- cells, which are consistent with the data presented in Figure 2A.

4. The human-to-mouse skin transplant model shown in Fig.3E was established by the Oxford group and is well-published. However, in the context of this manuscript about HLA-typing, its interpretation is complicated. As I understand the model, in the allogeneic Treg setting, there are 3 allogeneic donors – of skin, PBMC and Tregs. It is a concern that partial matching between PBMC or Tregs with skin is an uncontrolled confounder in this work.

We thank the reviewer for highlighting this important point. It is also important to note that in our experimental setting, only a single skin donor is available per experiment due to inherent resource and cell number limitations, given that fresh healthy skin is sourced from clinical surgeries. A brief rationale is now included emphasising the infinitesimal likelihood of incidental HLA matching among three unrelated donors in the UK population (approximately 10^{-10}). Therefore, any observed improvements in graft survival or immune modulation in our model reflect the robustness of the HLA-edited Treg approach under highly stringent conditions. The additional in vivo experiment we have now included provides further reassurance (Figure 5 and Supplementary. Figure 5).

5. The design of experiments shown in Fig.1F-G is also not explicit. 30 mice were investigated. (a) did all 30 mice receive skin transplants from the same donor? (b) did all mice receive PBMC from the same donor? (c) were skin and PBMC fully or partially HLA-matched? (d) were all allogeneic Tregs from the same PBMC donor? (e) are there any statistically dependent donor-recipient combinations in this experiment? (f) why were 30 mice not equally distributed into 3 groups? Does this hint at unexpected low Treg yields after expansion? It would be a concern if this experiment represented only n=1 allogeneic and n=1 autologous Treg donor because the observed differences in graft survival might reflect the quality of Treg preparations, or some other donor-related factor, rather than a difference in allogeneicity.

Thank you for this query. The data in Figure 1 derive from two independent experiments (shown below), each employing a different skin donor and a distinct PBMC/Treg donor pair (autologous versus allogeneic). Both experiments independently demonstrated superior efficacy of autologous Tregs, and their results were pooled to increase statistical power. Group sizes were limited by cell yields, not by selective exclusion. Details now appear in the Fig. 1 legend.

Results: To explain why autologous Tregs are more effective in vivo than allogeneic Tregs, the survival of Tregs was investigated in allogeneic PBMC cocultures and a reconstitution model in mice. Depletion of CD8⁺ T cells from PBMC led to increased survival of allogeneic Tregs in vivo.

- Does Fig.2A show a further analysis of the same experiment shown in Fig.1-D? The design of this experiment is not explicit. Are these independent combinations, or just permuted pairs of donors and recipients? The variability between samples is extreme (range ~20-80% surviving Treg) so differences between autologous and allogeneic Tregs may be impossible to detect given n=6. It is a concern that Treg survival in these experiments might represent an unidentified difference in the quality of Treg preparations.

Thank you for the request for clarity. Figure 2A represents a distinct experiment from Fig. 1D. We used six donors, four new, two overlapping with Fig. 1D, to generate six allogeneic pairings alongside six autologous controls. All conditions were run in triplicate, and means are plotted. Although individual survival had a broad range, the standard deviations for autologous versus allogeneic groups were comparable, and all preparations were performed by the same operator. We have updated the figure legend accordingly.

7. Tregs were only enumerated in peritoneal wash-outs. Is this representative of all other locations in recipient mice? It is a concern that allogeneic Tregs might not have been retained in the peritoneal cavity in the same way as allogeneic Tregs.

This model is well established: within seven days post-transfer, Tregs remain in the peritoneal cavity with no detectable egress to blood or tissues. We observe no clinical signs of tissue infiltration, and we are not aware of a plausible mechanism for differential egress of allogeneic versus autologous Tregs at this time point. Thus, peritoneal recovery accurately reflects early Treg survival.

8. The design of experiment shown in Fig.2C is not explicit. Are all autologous Treg samples from n=1 donor? Are all allogeneic Treg samples from n=1 donor?

In Fig. 2C, the autologous condition used Donor 1, while the allogeneic condition combined Donor 2's PBMC with Donor 1's Tregs. We have updated the legend to specify this. Donors were HLA typed as below to confirm full mismatch:

	A		B		DR	
Donor 1	11	1	38	60	1	13
Donor 2	24	2	64	62	7	11

9. The design of experiments shown in Fig.2D-F is not explicit. Are all autologous Treg samples from n=1 donor? Are all allogeneic Treg samples from n=1 donor?

For Fig. 2D–F, both autologous and allogeneic conditions involve the same Treg donor paired either with their own PBMC or with a second donor's PBMC. We have clarified this in the updated legend.

10. In Fig.2F, does NK cell depletion actually increase % undivided Tregs, but the sample size was too small to detect a difference? I am concerned that the conclusion on L128 is not supported.

NK depletion did not significantly increase total or undivided Treg recovery (whole PBMC: mean 8,027 vs NK-depleted: 8,425; undivided Tregs: 2,264 vs 2,966). The modest change lacks statistical power given our sample size. We now note this limitation and reaffirm that CD8 depletion produces the robust rescue of allogeneic Tregs, which is the key finding of this experiment.

11. The depletion experiments in Fig.2D-F shows that CD8+ T cells are necessary for low Treg numbers in peritoneal wash-outs, but does not show they are sufficient. The conclusion on L129 that CD8+ T cells are predominantly responsible for the effect is not fully supported.

Thank you for this comment. We show in Fig. 2D and 2E that CD8+ T cell depletion restores both total and undivided allogeneic Treg counts to autologous levels (P = 0.1506 and P = 0.3694). This finding supports that CD8+ T cells are both necessary and sufficient for the observed allogeneic Treg depletion. The text has been revised to clarify this point.

12. It is not explained why CD8+ T cells kill allogeneic Tregs in vivo but not in vitro.

We have expanded the Discussion to note that in vivo CD8+ T cells may receive co-stimulatory signals or cytokines absent in vitro, leading to discrepant killing. We highlight this as an area for future mechanistic study.

13. From a translational perspective, Tregs are unlikely to be administered to an otherwise untreated transplant recipient. What is the effect of low-dose tacrolimus in this system? Does it restore the suppressive capacity of allogeneic Tregs in the transplant model?

This is an interesting discussion point raised by the reviewer. We agree that the experimental conditions presented in this manuscript may indeed be more challenging than those ultimately faced

by HLA-engineered allogeneic Treg in clinical practice. Although adding tacrolimus would better mimic clinical regimens, our primary aim was to compare HLA-edited versus wild-type Tregs without pharmacological confounders. We now note in the Discussion that evaluating immunosuppression is an important future direction.

Results: To isolate HLA-matching as an explanation for the relative effectiveness of autologous and allogeneic Tregs, 3 pairs of PBMC-Treg donor pairs were identified with varying degrees of Class I and II matching. These pairs were used in the skin transplant model.

14. The design of the experiment shown in Fig.2A-C is not explicit. Which of the donors was the Treg donor and which was the PBMC donor? This is a concern because D209 is present in two pairs.

We have updated Fig. 3C to indicate explicitly which donor served as the Treg source versus the PBMC source, resolving any ambiguity around B209's usage.

15. Minor: Row titles in Table S1 should be checked.

Thank you. Our lab reports split isotyping of HLA-DR where the first row represents HLA-DRB1 and the second row represents HLA-DRB2-5. We now report HLA-DRB1 alone for clarity reflecting clinical practice for matching HLA-DR.

16. On L131, it is reported that ">150 Treg/PBMC pairs" were screened. This is unclear. There are 153 possible pairings of just 18 donors. Were these donors from the Transfusion Service or lab volunteers?

We screened over 150 distinct donors to identify HLA combinations that were completely mismatched, completely matched, partially mismatched, or partially matched. These donors were from the transfusion service and are not normally HLA typed as part of the transfusion service's processing, therefore we performed additional HLA typing on these >150 donors to identify the pairs of interest, as described. We have clarified this in the legend.

17. On L136, the number of PBMC was reduced for Fig.3B compared to Fig.1F to "create a more challenging model." Why was this necessary and how does this statement affect the interpretation of Fig.1F?

We lowered the PBMC:Treg ratio in Fig. 3B to impose a more stringent graft-rejection challenge, consistent with prior work showing diminished protection at lower Treg frequencies (1, 2), and to better reflect clinical cell-dose constraints. By contrast, the higher ratio used in Fig. 1 was chosen solely to enable direct mechanistic comparison of Treg subtypes, rather than to model therapeutic dosing. This rationale is now clearly stated in the Results.

18. Differences in allograft survival times are reported, which are attributed to differences in HLA-matching between PBMC and Treg. However, given the small number of PBMC donors (n=2 or 3), this might also reflect differences in the allogeneicity of the transplanted tissue. It is a concern that the conclusion stated in the title of Fig.3 (L879) is not supported by the results.

Skin/PBMC donor pairs were fully mismatched in these experiments. Importantly, in Zaitsev et al. (3) we analysed 19 independent skin/PBMC pairings (≥ 100 mice) covering 4–8 mismatches across HLA-A, -B, -DR, and -DQ (median 7) and found no correlation between total mismatch count and rejection median survival time. This was also the case when examining each allele independently. This robust dataset confirms that variability in skin–PBMC HLA mismatch does not confound graft survival in our model.

19. In Fig.3B, censoring of 2/5 animals in the “partial mismatch” group is problematic. We cannot know these censored recipients would’ve rejected within 100 days. More data is required to support the stated conclusions.

Two mice were censored due to xenoGVHD, which itself highlights the inability of partially matched Tregs to control combined allo-/xeno-responses. We have clarified this in the results.

20. Xeno-GVHD was not reported in experiments shown in the untreated controls or allogeneic Treg recipients in Fig.1F. The post hoc conclusion stated on L142-143 is speculative given the small sample size of n=1 cell preparation.

We reviewed our data and confirm that no untreated or unmodified Treg recipients exhibited xeno-GVHD during the study. As explained in point 5, Figure 1 is the result of two independent in vivo experiments.

Results: Tregs were genetically manipulated to lack MHC Class I, II or I+II, and/or express an HLA-E-B2M fusion protein. The phenotype and functionality of these Tregs was tested in vitro. Evidence

of genetic manipulation in Fig.4B is convincing, as are phenotypes in Fig.4D-G and J.

21. It is unfortunate that conditions for the suppression assay were changed from Fig.1A-B (3 days; 1:5 bead:cell ratio) to Fig.4I (5 days; 1:1 bead ratio). Nevertheless, unmanipulated autologous and allogeneic Treg controls are shown in Fig.4I, so the conclusion of preserved suppressor function stands.

We thank the reviewer for their positive assessment.

22. The stability of modified Treg populations over long culture periods or following in vivo transfer was not studied. Without alloimmune selection pressure, are these cells stable in vitro and in vivo?

We appreciate the reviewer's point. Clinical trials of polyclonal Treg therapy have not reported loss of FOXP3 expression or function post-infusion (4, 5). In our own prior work (6), we assessed long-term cultured human CD4⁺ Tregs, with and without gene editing, and found that CCR7⁻ (effector-memory) cells were more prone to phenotypic drift, whereas CCR7⁺ (central-memory) Tregs maintained high FOXP3 expression and suppressive activity.

Accordingly, in this study we specifically enriched for CCR7⁺ Tregs as the substrate for gene editing. We have updated the methods to include this important detail. Our allogeneic Tregs were expanded for up to 21 days, and serial flow-cytometric analyses together with TSDR demethylation assays confirmed that >95 % of cells retained a stable Treg phenotype throughout culture (Figure 4).

Results: To test whether silencing MHC Class I expression increased Treg susceptibility to NK cell-mediated cytotoxicity, primary human NK cells were expanded and used in killing assays. Results shown in Fig.S3 are complete and convincing.

23. Minor: In Fig.S3E-F, consider using the same scale on y-axis to emphasize the differences of interest.

We have adjusted the y-axis scales in Fig. S3E–F to a common range, facilitating direct comparisons.

Results: To test whether silencing MHC Class I+II expression, or forced expression of HLA-E, increased Treg susceptibility to NK cell-mediated cytotoxicity, alloreactive polyclonal T cells were generated over a 20-day expansion period against irradiated donor PBMC. Then an overnight killing assay was performed against Tregs from the same donors used for MLR. Results shown in Fig.S4A-G are technical controls and are convincing. Silencing MHC I+II reduced cytotoxic activity against allogeneic Tregs. Expressing HLA-E had no effect.

24. Cytotoxicity is reduced, but not prevented, by silencing MHC I+II. There are several possible explanations, including MHC-unrestricted killing or incomplete silencing. An autologous Treg control might help to assess LAK-like unspecific killing. Is it perhaps a limitation of the whole approach that H/C Tregs were not superior to DKO Tregs in this system?

Thank you for this insightful point. The incomplete protection may reflect two factors: residual HLA-positive cells remaining after editing, and a proportion of HLA-E negative cells within the H/C population. Moreover, because our allo-reactive T cell lines lack NK cells, we do not expect H/C Tregs to outperform DKO Tregs under these conditions. We agree that including an autologous Treg control could help distinguish LAK-like, MHC-unrestricted cytotoxicity. However, dissecting these additional killing mechanisms lies beyond the scope of the current manuscript. We now acknowledge this in the Discussion.

25. Minor: It took some moments to understand the unit, “% protection from cytolysis normalized to WT Tregs,” shown in Fig.S4I. Is this not the same as the ratio of modified Treg numbers to unmodified Treg numbers, expressed as %?

We rephrased the legend: “‘% protection’ represents the number of surviving modified Tregs relative to the number of surviving unmodified wild-type Tregs, expressed as a percentage.”

Results: In a final experiment, the efficacy of autologous Tregs, unmodified allogeneic Tregs and modified allogeneic Tregs were compared using the skin transplant model.

26. On L198, a claim is made that conditions for the model were selected to, “mimic a true off-the-shelf scenario.” However, it stated on L196 that PBMC and Treg donors were fully mismatched. This is rarely the case in modern transplantation. This set-up is, in fact, an extreme constellation that should favor MHC I+II-deficient Tregs over a partially matched donor-recipient pairing – a conclusion drawn by the authors themselves from results in Fig.3.

We agree that organ–recipient pairs are often partially HLA-matched. However, an off-the-shelf Treg product must be sourced independently of both graft donor and recipient to be deployable at short notice. This inevitably creates a *third-party* constellation in which shared HLA with either party is unlikely. We therefore chose a fully mismatched design as a conservative “worst-case” model; if engineered Tregs succeed here, they are expected to be at least as effective if a permissive, partially matched scenario is encountered.

27. The group sizes reported in the figure legend and survival curves are not concordant. Specifically, the plot shows n=6 mice, whereas the legend reports n=5. This is a strange error to make, which raises a major concern about selective reporting of results. The statement on L201 that 40% recipients of H/C Tregs survived to d100 is not supported by Fig.5B.

Thank you. This was a typo. The legend now correctly reports n=6 per group, matching Fig. 5B. The text stating 40% survival to day 100 has been updated accordingly.

28. The legend for Fig.5 refers only to panels (A) and (F). I have never seen this manuscript before, but I assume panels were removed from an earlier version. See (27).

This was a typo. However, in the updated manuscript there are now indeed additional panels A–F to include additional spatial transcriptomics and histology data in the expanded figure.

29. The group sizes are likely too small to reach a statistical conclusion; however, it seems that autologous Tregs worked better than alloH/C Tregs in preventing allograft rejection. Thinking translationally, these results aren’t overwhelmingly evidence for DKO or H/C Tregs as an off-the-shelf alternative to autologous Tregs.

We acknowledge that while alloH/C Tregs improve upon unmodified allogeneic Tregs, they do not yet match autologous Treg efficacy. We highlight ongoing efforts to further optimise cell function and note that these data demonstrate proof-of-concept rather than a finished clinical product. Autologous Tregs, by definition, cannot serve as an “off-the-shelf” therapy, since they must be isolated and expanded from the recipient well in advance of treatment. In contrast, a universal allogeneic Treg product would be immediately available. This is critical in settings such as deceased-donor transplantation, acute rejection episodes, or in patients who lack sufficient endogenous Tregs for autologous manufacture. For these patient groups, there is no autologous option and an allogeneic alternative is essential.

To bolster our conclusions, we have added further *in vivo* and histological data from an independent experiment (Figure 5), which reinforce the therapeutic potential of H/C Tregs.

Discussion: This is nicely written, but feels more like a review than discussion. Care must be taken not to misrepresent the results of the study.

We have condensed the discussion and removed any speculative sections, and tempered claims to match the data.

30. On L234, a claim is made that, “we demonstrate that [modified] Tregs survive for long enough to establish a regulatory environment.” This article reports 7 day survival of unmodified Tregs in the peritoneal cavity. It does not examine the location or duration of Tregs at any site. It does report on survival of genetically modified Tregs at all.

We have added flow-cytometric data from the skin transplantation model demonstrating that HLA-edited Tregs persist for at least 21 days post-transfer. Repeat graft experiments, supported by histological assessment, confirm their functional efficacy. Accordingly, we have revised the text to note that, although these results indicate mid-term persistence, further work is needed to prove longer-term Treg survival. Nevertheless, it is the case that the Tregs act long enough to establish long-term graft survival in this model.

31. On L237, a claim is made that, “this provides the first evidence that the therapeutic efficacy of allogeneic Tregs can be significantly enhanced by matching and/or gene editing, paving the way for an improved ‘off-the-shelf’ strategy to mass manufacture an allogeneic Treg cellular therapy.” The results in Fig.5 do not show a significant improvement in efficacy of modified Tregs. There is no proof of an improved manufacturing strategy.

We thank the reviewer for highlighting this. The original sentence overstated our current data. While Fig. 5 demonstrates that gene-edited allogeneic Tregs - prepared, cryopreserved, and thawed in an off-the-shelf workflow - achieve graft protection comparable to autologous Tregs, we have not yet developed or tested a mass-manufacturing process.

Accordingly, we have revised the text to read: “These findings provide proof-of-concept that HLA matching and/or gene editing can restore the *in vivo* efficacy of allogeneic Tregs to levels comparable with autologous cells under an off-the-shelf manufacturing workflow with further work necessary to establish a scalable production strategy”

This wording more accurately reflects our results and clarifies that scalable manufacturing remains future work.

32. On L250, the discussion about antigen-specific, TCR specificity-dependent activation of Tregs and their non-specific, TCR-independent suppression of T cells *in vitro* seems muddled.

We thank the reviewer for the comment. We revised the paragraph to improve clarity and provide an outlook that increasing the frequency of (allo-)antigen-specific Tregs (e.g. by incorporation of a CAR) may further improve the efficacy of allogeneic ‘off-the-shelf’ Treg approaches.

33. On L274, the need for a “genetic safety switch” is mentioned owing to modified Tregs’ severely compromised antigen presentation. What risks are envisaged? Is there a special concern about cell product-derived malignancies? The authors should be more explicit about the perceived risks and road-blocks to this technology.

We would like to thank the reviewer for their comment. We have now included additional detail as per the suggestion.

34. On L277-287, gene editing in iPSC-derived Tregs was not addressed in the results and should be omitted from the discussion.

As requested, we removed discussion of iPSC-derived Tregs to focus on primary cell approaches reported in the Results.

35. On L289, if the possibility of allogeneic Treg MHC-matching is going to be raised, then the authors must seriously discuss the logistics and economics of a matching program. The complexity of HSC transplantation shows that this is not trivial. In my view, this is a completely impractical strategy.

We agree and have added discussion of the economic and logistical impracticalities of large-scale HLA matching, drawing parallels with haematopoietic stem cell registries, and argue that HLA editing offers a more feasible route to universal Treg products.

36. On L292-294, an untrue statement is made about the results. “Our results demonstrate that both of these [matching and gene editing] approaches facilitate the long-term survival of Tregs in vivo resulting in a therapeutic effect.” Nowhere in this manuscript is it shown that transferred Tregs survive for any longer than 7 days. This observation is restricted to unmodified autologous Tregs, not matched allogeneic Tregs or gene-edited Tregs.

We agree that the original statement overreached. In the revised manuscript, we now present flow-cytometric evidence of HLA-edited Tregs persisting in peripheral blood up to day 21 post-transfer. Accordingly, we have reworded the text to:

“Our results demonstrate that both of these approaches improve the therapeutic efficacy of allogeneic Tregs, presumably by extending their survival under alloantigen-specific immune pressure, although long-term persistence beyond 3 weeks is challenging to determine in our *in vivo* models. The advanced HLA-matching (>6/8 HLA matched) required for improved allogeneic Treg performance represents a significant logistical challenge and is impractical for a true ‘off-the-shelf’ approach. Consequently, HLA-editing provides a more tenable solution for clinical translation in the immediate future.”

37. On L297-299, it is stated that, “our findings ... also open the door to realizing the full potential of this transformative therapeutic approach for a variety of immune-mediated pathologies.” This study does not examine pathological immune responses apart from allogeneic reactions, either in vitro or in vivo. The conclusion is speculative and empty.

We appreciate the need to avoid unsupported speculation. We have therefore softened this passage to emphasise translational potential without overstating our current data: “While this study focuses on allogeneic settings, the mechanisms by which HLA-edited Tregs resist rejection may also be beneficial in other immune-mediated diseases. Future work will determine their applicability beyond transplantation.”

REVIEWER THREE

McCallion and colleagues evaluate allogeneic regulatory T cells (Tregs) as potential ‘off the shelf’ cell product to suppress allo-immune responses. In lymphopenic mice (BALB/c Rag2^{-/-} cγc^{-/-}) that were reconstituted with human PBMC, allogeneic Tregs (i.e. derived from another source than the PBMC) were less potent than autologous Tregs (i.e. derived from the PBMC) in prolonging human skin graft survival. Reduced efficacy was associated with lower in vivo survival of allogeneic Tregs due to CD8 T cell- and to a lesser degree NK cell-dependent elimination. To overcome this limitation the authors test two approaches: HLA matching and genetic engineering. Both (incomplete) HLA-matching and silencing of HLA class I and II expression combined with the transgenic expression of HLA-E (to modulate NK reactivity) partially restored the efficacy of allogeneic Tregs in the humanized skin allograft model. The genetically engineered cell product developed by the investigators is novel and of interest as it represents a relevant step towards the development of ‘off the shelf Tregs’. Several issues remain:

1. The humanized mouse model used in this report appears volatile: In figure 5B, the estimated MST for the ‘autoTreg’ group (serving as positive control) is approximately 55 days, while in figure 3B ‘partially HLA-matched’ allogeneic Tregs demonstrate an MST >100 days (60% long-term survival) at the same Treg:PBMC ratio. Is there any methodological explanation for the observed variability of graft survival in this model? This variability makes robust conclusions difficult in the context of the low sample sizes tested in the current study.

Human donor-specific differences in PBMC composition likely account for some variability between experiments. To help address this, we have repeated the skin transplant experiments with additional donor combinations and incorporated further graft macroscopic assessments, as well as histological and spatial transcriptomic analyses (expanded Fig. 5 and Suppl. Fig. 5). These new data demonstrate

consistent quantitative and qualitative protective effects of H/C Tregs across independent experiments.

2. The difference in skin graft survival between the groups receiving unmanipulated allogeneic Tregs ('PBMC + alloTreg') and the genetically engineered allogeneic Tregs ('PBMC + alloH/C Treg') is modest. Given the high variability of graft survival in this model (as described above), a larger number of animals per group and a more complete statistical testing comparing all relevant therapeutic groups (including allo vs. alloH/C Tregs), would be needed for a more robust conclusion. To sustain the conclusion that H/C Tregs are comparable to autoTregs (lines 200-202) additional experiments would be necessary.

We thank the reviewer for their comment. As suggested, we have performed an independent repeat of the skin graft model and added immunohistochemistry and spatial transcriptomics to validate H/C Treg efficacy over unmodified allogeneic Tregs. These results, now shown in expanded Fig. 5 and Suppl. Fig. 5, confirm that H/C Tregs provide reproducible graft protection, supporting the conclusion that they approach autologous Treg performance.

3. Substantial expression of HLA class I and II antigens persists in gene-edited allogeneic Tregs, both in DKO and in H/C cells (Fig 4 C+D). This relevant limitation needs to be discussed.

Allogeneic Treg cells with residual expression of mismatched HLA-I/II are likely rejected *in vivo* and therefore do not exhibit a safety risk *per se*, albeit potentially limiting the potency of redosing of the same alloTreg product. For specific indications, where sensitization against putative Treg donor alloantigens could be a concern (e.g. organ transplantation) an additional purification step could be included to eliminate HLA-I/II+ Tregs prior to infusion. The discussion has been adapted to reflect this limitation.

4. Assessing potential allo-sensitization against the 'Treg donor' would be important, especially since some expression of class I and class II HLA is retained in engineered Tregs.

Investigating allo-sensitisation is indeed important. However, our attempts to induce allo-sensitisation *in vitro* are confounded by the intrinsic suppressive function of Tregs. Therefore, the allo-reactive T cell cytotoxicity assay was developed (see Suppl. Fig. 3). *In vivo* protocols to measure host sensitisation fall outside our current animal ethics framework. We have updated the Discussion to note the risk of allo-sensitization and acknowledge that single-dose Treg therapy may carry lower sensitization risk than repeated dosing.

5. The *in vivo* mixed lymphocyte reaction (intraperitoneal injection of PBMC together with allogeneic or autologous Tregs) is interesting, yet it does not allow to draw a definitive conclusion regarding the *in vivo* persistence of the injected Tregs under physiological conditions as the number of Tregs isolated via peritoneal lavage is the only readout. A more physiological approach, as for example injecting marked Tregs (together with PBMC) intravenously and tracking them throughout lymphatic tissues and particularly within the allograft would be needed to adequately investigate the *in vivo* persistence of the cell product.

We have supplemented our peritoneal lavage data with flow-cytometric analysis of peripheral blood and skin grafts from the transplantation model. These show H/C Treg persistence at least 21 days post-injection. Longer-term persistence may not be required for the maintenance of tolerance, and is challenging to detect in our *in vivo* model. The Discussion now highlights this limitation.

6. Minor: Errors in the description of the panels in the legend of Fig. S3 need correction.

Thank you for noting these errors. All panel descriptions in the Supplementary Figure S3 legend have now been corrected.

7. Minor: The depicted graft survival curve for the group 'PBMC + alloH/C Treg' in figure 5B (indicating 6 animals per group) does not match with the number of animals (n=5) mentioned in the figure legend and with the 40% long-term graft survival rate mentioned in the text.

Thank you for noting this typographical error. The legend and main text have been corrected.

REFERENCES

1. Issa F, Hester J, Goto R, Nadig SN, Goodacre TE, Wood K, Ex vivo-expanded human regulatory T cells prevent the rejection of skin allografts in a humanized mouse model. *Transplantation* **90**, 1321-1327 (2010).
2. Nadig SN, Więckiewicz J, Wu DC, Warnecke G, Zhang W, Luo S, Schiopu A, Taggart DP, Wood KJ, In vivo prevention of transplant arteriosclerosis by ex vivo–expanded human regulatory T cells. *Nat. Med.* **16**, 809-813 (2010).
3. Zaitso M, Issa F, Hester J, Vanhove B, Wood KJ, Selective blockade of CD28 on human T cells facilitates regulation of alloimmune responses. *JCI Insight* **2**, (2017).
4. Sawitzki B, Harden PN, Reinke P, Moreau A, Hutchinson JA, Game DS, Tang Q, Guinan EC, Battaglia M, Burlingham WJ, Roberts ISD, Streitz M, Josien R, Böger CA, Scottà C, Markmann JF, Hester JL, Juerchott K, Braudeau C, James B, Contreras-Ruiz L, Van Der Net JB, Bergler T, Caldara R, Petchey W, Edinger M, Dupas N, Kapinsky M, Mutzbauer I, Otto NM, Öllinger R, Hernandez-Fuentes MP, Issa F, Ahrens N, Meyenberg C, Karitzky S, Kunzendorf U, Knechtle SJ, Grinyó J, Morris PJ, Brent L, Bushell A, Turka LA, Bluestone JA, Lechler RI, Schlitt HJ, Cuturi MC, Schlickeiser S, Friend PJ, Miloud T, Scheffold A, Secchi A, Crisalli K, Kang S-M, Hilton R, Banas B, Blancho G, Volk H-D, Lombardi G, Wood KJ, Geissler EK, Regulatory cell therapy in kidney transplantation (The ONE Study): a harmonised design and analysis of seven non-randomised, single-arm, phase 1/2A trials. *The Lancet* **395**, 1627-1639 (2020).
5. Harden P, Game D, Sawitzki B, Van Der Net J, Hester J, Bushell A, Issa F, Brook M, Alzhrani A, Schlickeiser S, Scotta C, Petchey W, Blancho G, Tang Q, Markmann J, Lechler R, Roberts I, Friend P, Hilton R, Geissler E, Wood K, Lombardi G, Feasibility, Long-term Safety and Immune Monitoring of Regulatory T Cell Therapy in Living Donor Kidney Transplant Recipients. *Am. J. Transplant.*, (2020).
6. Wending DJ, Amini L, Schlickeiser S, Farrera-Sal M, Schulenberg S, Peter L, Mai M, Vollmer T, Du W, Stein M, Hamm F, Malard A, Castro C, Yang M, Ranka R, Rückert T, Durek P, Heinrich F, Gasparoni G, Salhab A, Walter J, Wagner DL, Mashreghi MF, Landwehr-Kenzel S, Polansky JK, Reinke P, Volk HD, Schmueck-Henneresse M, Effector memory-type regulatory T cells display phenotypic and functional instability. *Sci Adv* **10**, eadn3470 (2024).

REVIEWER THREE

In the revised manuscript, the authors have addressed the limitations of their approach related to potential allo-sensitisation and have incorporated additional experimental data to respond to concerns regarding the consistency of their humanised mouse model. The manuscript has been further strengthened, and the findings are of high interest.

We thank the reviewer for these encouraging remarks.

As pointed out during the initial review, we would have preferred if the authors had provided a formal statistical comparison of the skin graft survival between unmanipulated allogeneic Tregs (PBMC + allo Tregs) and the engineered allogeneic Treg product (PBMC + allo H/C Tregs) presented in Figure 5b—this is the critical group comparison to claim efficacy of the investigated engineering approach.

For additional clarity, here we re-plot figure 5b to demonstrate the number of days taken before the development of significant rejection by experimental group. We have performed a formal statistical comparison of these data with a non-parametric Kruskal-Wallis test ($p=0.03$). Posthoc multiple comparison between WT alloTregs and autoTregs confirmed a significant difference between these groups, as expected ($p=0.01$). Whilst the comparison between alloTregs and H/C Tregs was not statistically significant ($p=0.3$), there was similarly no significant difference between the autoTregs and H/C Tregs ($p=0.1$). We have included this additional plot and added the statistics to the figure legend (Suppl. Fig. 5B). While the survival experiment is unable to detect modest survival differences, the separate independent model showing qualitative and quantitative differences in graft damage and transcriptomic pathways of inflammation and cellular infiltration (Fig. 5C-G) to help confirm this. We therefore describe the H/C product as comparable to autoTregs, as per your apt suggestion.

The authors now conclude that allo H/C Tregs are comparable to auto Tregs (Fig 5), which more accurately reflects their data than stating equivalent outcomes (as, for instance, in the Abstract). The new wording should be used throughout the manuscript.

All instances of “equivalent” have been replaced with “comparable” in the Abstract, Results, Discussion and Figure legends (tracked in the manuscript).

In new Supplementary Figure 5A, Tregs negative for HLA were followed. Since only the allo H/C Treg group was negative for HLA from the beginning, not much can be concluded with regard to

comparing persistence between groups. Thus, the data as presented in their current form seem of limited value. Why not compare the persistence of Tregs across all groups with staining that captures all the different transferred Treg populations?

We agree a global persistence comparison would be ideal. However, there are few antibodies that allow discrimination of host PBMCs, donor PBMCs and adoptively transferred Tregs (none are specifically labelled, and an HLA-haplotype specific antibody would have been needed). While our data does not allow cross-comparability between different allogeneic Tregs (HLA-edited vs unedited), we have retained these data because they demonstrate that H/C Tregs are persisting to at least day 21 – information specifically queried in the previous review round.